# In-Context Learning from Training on Unstructured Data: The Role of Co-Occurrence, Positional Information, and Training Data Structure

**Kevin Christian Wibisono** [1]   **Yixin Wang** [1]

## Abstract

Large language models (LLMs) like transformers have impressive in-context learning (ICL) capabilities; they can generate predictions for new queries based on input-output sequences in prompts without parameter updates. While many theories have attempted to explain ICL, they often focus on structured training data similar to ICL tasks, such as regression. In practice, however, these models are trained in an unsupervised manner on unstructured text data, which bears little resemblance to ICL tasks. To this end, we investigate how ICL occurs from unsupervised training on unstructured data. The key observation is that ICL can arise simply by modeling co-occurrence information using classical language models like continuous bag of words (CBOW), which we theoretically prove and empirically validate. Furthermore, we establish the necessity of positional information and nuisance token structure to generalize ICL to unseen data. Finally, we present instances where ICL fails and provide theoretical explanations; they suggest that the ICL ability of LLMs to identify certain tasks can be sensitive to the structure of the training data.

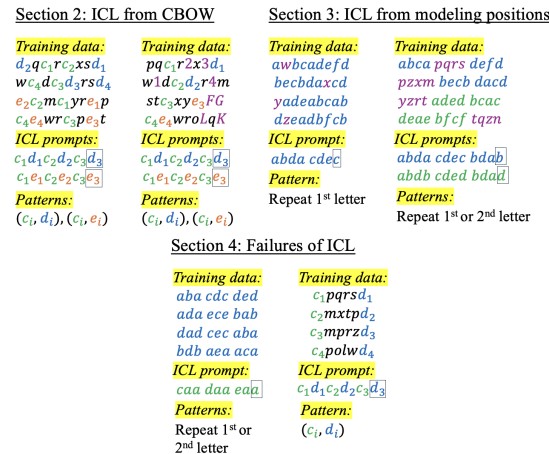

*Figure 1.* This paper aims to understand how in-context learning (ICL) occurs from pretraining on unstructured natural language data. In Section 2, we show that ICL can arise merely through modeling co-occurrence information using continuous bag of words (CBOW). Violet represents relationship-specific nuisance tokens. In Section 3, we establish the necessity of positional information and blocked nuisance structure for certain ICL tasks. Violet represents nuisance tokens. In Section 4, we present two scenarios where ICL can fail and provide theoretical explanations, highlighting the importance of training data structure in enabling the ICL ability of language models. Boxed letters represent the expected outputs. In the failed scenarios, the model predicts $e$ and $d_1$.

## 1. Introduction

Large language models (LLMs) such as transformers demonstrate impressive in-context learning (ICL) abilities (Brown et al., 2020): without updating parameters, they can identify tasks and generate predictions from prompts containing input-output examples. For example, given the prompt *dog anjing, cat kucing, lion singa, elephant*, a well-trained LLM should detect the English-to-Indonesian pattern and predict *gajah*, the Indonesian translation for *elephant*, as the most

[1]Department of Statistics, University of Michigan, Ann Arbor, MI, USA. Correspondence to: Kevin Christian Wibisono <kwib@umich.edu>.

*Proceedings of the 1ˢᵗ Workshop on In-Context Learning at the 41ˢᵗ International Conference on Machine Learning*, Vienna, Austria. 2024. Copyright 2024 by the author(s).

likely next token. The ICL ability of LLMs is surprising for at least two reasons. First, LLMs are trained from unstructured natural language data in an unsupervised manner via next-token prediction. Second, the training data of LLMs likely does not include sentences that resemble typical ICL prompts, i.e., of the form $c_1 d_1 \cdots c_K d_K$, where $(c_k, d_k)$ represents a known input-output pair.

Many efforts have sought to understand ICL from various theoretical and empirical perspectives; see related work in Section A. Some studies (e.g., Akyürek et al. (2022); Von Oswald et al. (2023); Dai et al. (2023); Zhang et al. (2024); Ahn et al. (2024)) expanded Garg et al.'s [2022] regression formulation and attributed transformers' ICL ability to gradient descent. Other studies (e.g., Wang et al. (2023), Zhang et al. (2023), Chiang and Yogatama (2024))

adopted a Bayesian perspective, building upon Xie et al.'s [2021] argument that ICL performs implicit Bayesian inference. While these connections are theoretically intriguing, they do not fully capture the actual ICL phenomenon: ICL arises from training on unstructured natural language data that are distinct from ICL prompts.

**This work.** We study *how ICL arises from pretraining on unstructured natural language data*. Throughout the paper, we focus on two types of ICL tasks. The first type involves known input-output pairings that frequently occur together in a sentence, such as *(country)-(capital)* and *(English word)-(Indonesian translation)*. The second type involves recognizable patterns that may not commonly co-occur in a sentence, such as *(word)-(first letter)*.

For the first task (left of Fig. 1), we examine cases where the training sentences contain one or two distinct input-output relationship types. We also study more realistic scenarios where some input-output pairs do not always co-occur and two types of relationships can co-occur in a single sentence. We prove that, in most cases, *ICL is achievable by only modeling co-occurrence information using continuous bag of words (CBOW)* (Mikolov et al., 2013), a pre-transformer language model. Further, we conduct prompting and synthetic data experiments to support our theoretical findings.

For the second task (right of Fig. 1), we study cases where the training sentences contain one or two distinct patterns, and a more realistic scenario where nuisance tokens are present. We prove that *positional information and blocked nuisance structure* (e.g., *pqrs* in Figure 1) *are crucial for ICL*. This observation aligns with Chen et al.'s [2024b] empirical finding that parallel structures in pretraining data support ICL. Moreover, we find that learned positional embeddings generally perform better, except in noisy scenarios where the nuisance tokens are not clustered in blocks.

Finally, we present two scenarios where ICL can fail regardless of model architectures (bottom of Fig. 1). In the first scenario (left example), both the training data and test prompts follow repeating patterns across blocks, but the pattern being repeated in the test data differs from that in the training data. In the second scenario (right example), training sentences contain known input-output pairs but only at fixed locations. These findings, along with their empirical and theoretical explanations, show that *LLMs may require specific pretraining data structures to exhibit ICL ability*.

**Summary of contributions.** In this paper, we (1) theoretically and empirically show that ICL can arise from merely modeling *co-occurrence patterns* using CBOW, (2) prove that, in other instances, ICL requires modeling *positional information* and *blocked nuisance structure*, and (3) present scenarios where ICL fails, highlighting the crucial role of *training data structure* for ICL to arise.

## 2. In-context learning can arise by merely modeling co-occurrence via CBOW

In this section, we focus on in-context learning (ICL) tasks involving pairings that commonly co-occur within training sentences. To motivate our discussion, we revisit the (English word)-(Indonesian translation) example in Section 1. Below we perform a simple experiment with ChatGPT 3.5 (OpenAI, 2022). The model is given prompts of the following form:

```
Provide the most plausible next token
to complete this sentence (only the
answer). Even if the sentence does not
make sense, please complete it as best
as you can: dog anjing, cat kucing,
lion singa, [word]
```

We take turns replacing [word] with elephant, tiger, soon, and main. For the first two options, ChatGPT 3.5 correctly outputs gajah and harimau, their respective Indonesian translations. However, it does not provide the correct outputs for the latter two: it follows soon with lebih baik beri makanan haiwan! (*better feed the animals!*) and main with bola (*ball*).[1] A similar pattern is observed with LLaMA 2 (Touvron et al., 2023), which produces the correct translations the first two words but incorrectly continues the last two words with to-be-published and an[1], respectively.

If ICL stems from the ability of LLMs to recognize consistent mappings in test prompts, these models should be equally likely to produce the correct answer for any given *[word]*, irrespective of its relevance to the in-context examples. However, this experiment demonstrates that this is not the case; in Section 2.4, we also present two similar experiments on countries, US states, and their capital cities. This naturally raises the question: *Can ICL arise from modeling co-occurrence information using a simple model like continuous bag of words (CBOW)* (Mikolov et al., 2013)?

**ICL via CBOW.** We prove that, for certain tasks, ICL is achievable by modeling co-occurrence information between pairs of tokens (regardless of their positions) using CBOW. We *do not* prove that ICL in transformer-based models arises through learning co-occurrence patterns. We utilize a variant of CBOW where each center word is modeled conditional on all other words in a sentence, rather than just neighboring words. Specifically, we associate each word $w$ with their center and context embeddings $u_w$ and $v_w$ of the same dimension. Given a sentence $x_1 x_2 \cdots x_I$, the $i$-th word ($x_i$) is distributed conditional on the other words in the sentence ($x_{-i}$): $p(x_i = k \mid x_{-i}) \propto \exp\left((u_k^\top \sum_{j \neq i} v_{x_j})/(I-1)\right)$.

---

[1] In Indonesian, main means *play*, main bola means *play soccer*, and mainan means *toy*.

The $u_w$'s and $v_w$'s are learned by minimizing the sum of the cross-entropy losses across all sentences and positions.

**Roadmap of Section 2.** In Section 2.1, we consider a simple ICL task of the form $c_{i_1}d_{i_1} \cdots c_{i_\ell}d_{i_\ell}c_{i_{\ell+1}}$, where $(c_i, d_i)$ represents a known pairing (e.g., a country and its capital city) and $i_1, i_2, \cdots, i_{\ell+1}$ are all distinct. The focus is to investigate whether a trained CBOW model can correctly output $d_{i_\ell}$. We also explore two other scenarios: ICL tasks of the form $c_{i_1}d_{i_1} \cdots c_{i_\ell}d_{i_\ell}c_{i_{\ell+1}}$ and $c_{i_1}e_{i_1} \cdots c_{i_\ell}e_{i_\ell}c_{i_{\ell+1}}$ in Section 2.2 (two connected relationships), as well as $c_{i_1}d_{i_1} \cdots c_{i_{\ell-1}}d_{i_{\ell-1}}c_{i_\ell}$ and $e_{i_1}f_{i_1} \cdots e_{i_\ell}f_{i_\ell}e_{i_{\ell+1}}$ (two disconnected relationships) in Section 2.3. Sections 2.4 and 2.5 conclude with prompting and synthetic data experiments that provide support to the theory.

## 2.1. ICL on single-relationship tasks

We investigate ICL in single-relationship tasks that take the form of $c_{i_1}d_{i_1} \cdots c_{i_\ell}d_{i_\ell}c_{i_{\ell+1}}$, where $(c_i, d_i)$'s denote known pairings such as countries and their capital cities. The vocabulary consists of $c_{1:K}, d_{1:K}, r_{1:L}$, where $r'_is$ represent other words (e.g., stop words). We first introduce Theorem 2.1, which states that ICL can arise if each sentence consists of exactly one $(c_i, d_i)$ pair, as long as the number of in-context examples ($\ell$) is not too large. To simplify calculations, we replace the cross-entropy loss with the squared loss. This involves removing the softmax activation and comparing the outputs against the one-hot encoding of the target words. The proof of Theorem 2.1 is in Appendix B.

**Theorem 2.1** (ICL on single-relationship tasks). *Let $K, L \geq S \geq 3$. Suppose each training sentence is generated by selecting one $(c_i, d_i)$ pair and $S - 2$ distinct $r_i$'s uniformly at random. We train a CBOW model with the squared loss and a sufficiently large embedding dimension on these sentences. Given a prompt $c_{i_1}d_{i_1} \cdots c_{i_\ell}d_{i_\ell}c_{i_{\ell+1}}$ with distinct $i_k$'s, the model correctly predicts $d_{i_{\ell+1}}$ if and only if $2\ell + 1 < \frac{KL(S-1)^3}{(K+L)(S-2)^2(S-1)+K(S-2)(S-1)^2-2(S-2)^4}$.*

As an example, when each training sentence contains exactly one *country-capital* pair (i.e., $(c_i, d_i)$), Theorem 2.1 says a trained CBOW model will correctly predict $d_{i_{\ell+1}}$ (i.e., the capital city of $c_{i_{\ell+1}}$) given an ICL prompt of the form $c_{i_1}d_{i_1} \cdots c_{i_\ell}d_{i_\ell}c_{i_{\ell+1}}$, provided that the prompt length ($2\ell + 1$) is not too large. This behavior is intuitively due to the presence of $c_{i_{\ell+1}}$ in the ICL prompt, leading the model to correctly predict $d_{i_{\ell+1}}$ due to the frequent occurrences of the pair $(c_{i_{\ell+1}}, d_{i_{\ell+1}})$ in the training data. However, when the prompt length is too large, the model will instead predict one of the $r_i$'s (see Theorem 2.1's proof in Appendix B for more details). If we let $L \to \infty$ and fix $K$ and $S$, the condition in Theorem 2.1 becomes $2\ell + 1 < K(S-1)^2/(S-2)^2$. This inequality trivially holds if the prompt length is set to be $S - 1$ to match the training sentences.

It is possible to adapt the proof of Theorem 2.1 to han-

*Table 1.* ICL on various single-relationship tasks, averaged over 10 repetitions, demonstrates stable, good performance across embedding dimensions ($d_E$), as Theorem 2.1 suggests. The corrupted setting also shows excellent ICL ability under certain scenarios.

| | Clean | | Corrupted | |
|---|---|---|---|---|
| $(p_0, p_1, p_2)$ | $d_E = 10$ | $d_E = 100$ | $d_E = 10$ | $d_E = 100$ |
| $(0, 1, 0)$ | 0 | 0 | 0 | 0 |
| $(0, 0, 1)$ | 0 | 0 | 0 | 0 |
| $(1/2, 1/2, 0)$ | 1 | 0.99 | 0 | 0 |
| $(1/2, 0, 1/2)$ | 1 | 1 | 1 | 1 |
| $(0, 1/2, 1/2)$ | 1 | 1 | 0 | 0.01 |
| $(1/3, 1/3, 1/3)$ | 1 | 1 | 1 | 1 |

dle the case when each sentence comprises exactly two (not one) different $(c_i, d_i)$ pairs. In this case, letting $L \to \infty$ and fixing $K$ and $S$, the model correctly predicts $d_{i_{\ell+1}}$ given the same ICL prompt if and only if $2\ell + 1 < \frac{K(K-2)(S-1)^2}{(K-2)(S-2)(S-4)-K}$. This upper bound is strictly larger than $K(S-1)^2/(S-2)^2$: when each sentence contains exactly two $(c_i, d_i)$ pairs, ICL under the squared loss occurs for longer prompts.

**Experiments.** To verify Theorem 2.1 and its generalizations, we conduct experiments using the cross-entropy loss with $S = 8$, $K = 10$, $L = 20$, and $\ell = 3$. We explore multiple $(p_0, p_1, p_2)$ values, where $p_k$ denotes the probability of having exactly $k$ pairs of $(c_i, d_i)$ in the sentence. For each $(p_0, p_1, p_2)$ triple, we introduce a more realistic setting where $c_i$ and $d_i$ do not always appear together by considering its *corrupted* version. In this setup, each $(c_i, d_i)$ pair has a 25% chance of being replaced with $(c_i, r_j)$ and a 25% chance of being replaced with $(d_i, r_j)$, for some $j \in [L]$.

Table 1 displays the average accuracy for each scenario, calculated over 10 repetitions. Notably, when $(p_0, p_1, p_2)$ is $(0, 1, 0)$ or $(0, 0, 1)$, ICL under the cross-entropy loss achieves zero accuracy, in contrast to perfect accuracy with the squared loss as shown in Theorem 2.1. We believe this difference in accuracy is an artifact of the loss functions used, although its relevance is limited by the fact that, in reality, it is unlikely for every sentence to contain at least one $(c_i, d_i)$ pair. On the other hand, perfect ICL performance is observed in other settings (e.g., when the training sentences contain either zero, one, or two $(c_i, d_i)$ pairs) in both the clean and corrupted scenarios. For an in-depth comparison of ICL performance using both the squared and cross-entropy loss across various numbers of demonstration examples, refer to Appendix C.

## 2.2. ICL on dual-connected-relationship tasks

In Section 2.1, we discussed the case where the training sentences contain one relationship, namely $(c_i, d_i)$'s. We now explore ICL on dual-connected-relationship tasks,

*Table 2.* ICL on dual-*connected*-relationship tasks, averaged over 10 repetitions, achieves perfect accuracy when $(p_0, p_1, p_2) \in \{(1/2, 0/1, 2), (0, 1/2, 1/2), (1/3, 1/3, 1/3)\}$ regardless of architectures and embedding dimensions $(d_E)$, as Theorem 2.2 suggests. When $(p_0, p_1, p_2) = (1/2, 1/2, 0)$, ICL performs better under imbalanced or extreme scenarios and with larger $d_E$.

| | Balanced | | Imbalanced | | Extreme | |
|---|---|---|---|---|---|---|
| $(p_0, p_1, p_2)$ | $d_E = 10$ | $d_E = 100$ | $d_E = 10$ | $d_E = 100$ | $d_E = 10$ | $d_E = 100$ |
| $(0, 1, 0)$ | $(0, 0)$ | $(0, 0)$ | $(0, 0)$ | $(0, 0)$ | $(0, 0)$ | $(0, 0)$ |
| $(0, 0, 1)$ | $(0, 0)$ | $(0, 0)$ | $(0, 0)$ | $(0, 0)$ | $(0.07, 0.10)$ | $(0, 0)$ |
| $(1/2, 1/2, 0)$ | $(0.53, 0.47)$ | $(0.51, 0.50)$ | $(0.69, 0.68)$ | $(1, 1)$ | $(0.94, 0.93)$ | $(1, 1)$ |
| $(1/2, 0, 1/2)$ | $(1, 1)$ | $(1, 1)$ | $(1, 1)$ | $(1, 1)$ | $(1, 1)$ | $(1, 1)$ |
| $(0, 1/2, 1/2)$ | $(1, 1)$ | $(1, 1)$ | $(1, 1)$ | $(1, 1)$ | $(1, 1)$ | $(1, 1)$ |
| $(1/3, 1/3, 1/3)$ | $(1, 1)$ | $(1, 1)$ | $(1, 1)$ | $(1, 1)$ | $(1, 1)$ | $(1, 1)$ |

where the two types of relationships are connected and denoted by $(c_i, d_i)$ and $(c_i, e_i)$: $c_i$ might represent a country, $d_i$ its capital city, and $e_i$ its currency. Our vocabulary comprises $c_{1:K}, d_{1:K}, e_{1:K}, r_{1:L}$, where $r_i$'s represent other words. The corresponding ICL tasks thus take the form $c_{i_1} d_{i_1} \cdots c_{i_\ell} d_{i_\ell} c_{i_{\ell+1}}$ and $c_{i_1} e_{i_1} \cdots c_{i_\ell} e_{i_\ell} c_{i_{\ell+1}}$, where the model is expected to output $d_{i_{\ell+1}}$ and $e_{i_{\ell+1}}$. This involves *task selection* as the model should use the in-context examples to infer the task. We first present Theorem 2.2, which states that a trained CBOW model can perform task selection if each sentence contains exactly two distinct $(c_i, d_i)$ pairs or two distinct $(c_i, e_i)$ pairs with uniform probability. Its proof is in Appendix D. While we can also theoretically show that ICL works in this case (up to a certain number of training examples), the calculations are extremely tedious. Thus, we only present empirical evidence in Table 2.

**Theorem 2.2** (Task selection in CBOW). *Let $K, L \geq 2$ and $S \geq 5$. Suppose each training sentence is generated by selecting two distinct $(c_i, d_i)$ pairs or $(c_i, e_i)$ pairs and $S - 4$ distinct $r_i$'s uniformly at random. We train a CBOW model with the squared loss and a large enough embedding dimension. Given a prompt $c_{i_1} d_{i_1} \cdots c_{i_\ell} d_{i_\ell} c_{i_{\ell+1}}$ $(c_{i_1} e_{i_1} \cdots c_{i_\ell} e_{i_\ell} c_{i_{\ell+1}})$ with distinct $i_k$'s, the model is more likely to predict $d_{i_{\ell+1}}$ $(e_{i_{\ell+1}})$ than $e_{i_{\ell+1}}$ $(d_{i_{\ell+1}})$.*

Theorem 2.2 says that, when each training sentence includes two $(c_i, d_i)$ pairs or two $(c_i, e_i)$ pairs, a trained CBOW model is capable of task selection. To understand this result, consider the ICL prompt of the first type, i.e., $c_{i_1} d_{i_1} \cdots c_{i_\ell} d_{i_\ell} c_{i_{\ell+1}}$. Here, the output is more likely to be $d_{i_{\ell+1}}$ than $e_{i_{\ell+1}}$ since $d_{i_{\ell+1}}$ co-occurs with the other $d_{i_j}$'s in the training data (and $e_{i_{\ell+1}}$ does not). In Theorem 2.2, we unrealistically require each sentence to contain either two distinct $(c_i, d_i)$ pairs or $(c_i, e_i)$ pairs. However, this condition is not necessary as we empirically show next.

**Experiments.** We use the cross-entropy loss with $S = 8$, $K = 10$, $L = 60$, and $\ell = 3$. Each training sentence is equally likely to be a *cd* sentence (i.e., containing $(c_i, d_i)$ pairs) or a *ce* sentence (i.e., containing $(c_i, e_i)$ pairs), but not both. We explore multiple $(p_0, p_1, p_2)$'s, where $p_k$ is

the probability of having exactly $k$ pairs of $(c_i, d_i)$ for a *cd* sentence, or $k$ pairs of $(c_i, e_i)$ for a *ce* sentence.

Additionally, we introduce three different scenarios: *balanced*, where all $L$ random words are equally likely to occur in both *cd* and *ce* sentences; *imbalanced*, where $L/3$ words are more likely to occur in *cd* (*ce*) sentences; and *extreme*, where $L/3$ of the words can only occur in *cd* (*ce*) sentences. Table 2 shows the accuracies of both tasks for each scenario, averaged over 10 repetitions. We observe a perfect accuracy when $(p_0, p_1, p_2) \in \{(1/2, 0/1, 2), (0, 1/2, 1/2), (1/3, 1/3, 1/3)\}$ across all embedding dimensions and scenario types. The near-zero accuracy when $(p_0, p_1, p_2)$ or $(0, 1, 0)$ or $(0, 0, 1)$ is again an artifact of the cross-entropy loss discussed in Section 2.1.

Interestingly, ICL works in the imbalanced and extreme scenarios when $(p_0, p_1, p_2) = (1/2, 1/2, 0)$, where sentences do not contain more than one $(c_i, d_i)$ or $(c_i, e_i)$ pair. To see this, consider the balanced scenario where each $r_i$ is equally probable to appear in both types of sentences. Given a prompt of the form $c_{i_1} d_{i_1} \cdots c_{i_\ell} d_{i_\ell} c_{i_{\ell+1}}$, it is easy to see that the model should output $d_{i_{\ell+1}}$ or $e_{i_{\ell+1}}$ with equal probability. On the other hand, in the imbalanced and extreme scenarios, the signals from the $r_i$'s can allow for task selection, thus contributing to the success of ICL.

### 2.3. ICL on dual-disconnected-relationship tasks

We next replicate the experiments in Section 2.2, but with two disconnected relationships $(c_i, d_i)$ and $(e_i, f_i)$. For example, $(c_i, d_i)$ might represent a country and its capital city and $(e_i, f_i)$ might represent a company and its CEO. Our vocabulary consists of $c_{1:K}, d_{1:K}, e_{1:K}, f_{1:K}, r_{1:L}$, where $r_i$'s represent other words. Table 3 summarizes the accuracies of the ICL tasks $c_{i_1} d_{i_1} \cdots c_{i_\ell} d_{i_\ell} c_{i_{\ell+1}}$ and $e_{i_1} f_{i_1} \cdots e_{i_\ell} f_{i_\ell} e_{i_{\ell+1}}$ for each scenario, averaged over 10 repetitions. Similar to the connected setting in Section 2.2, we observe a perfect accuracy when $(p_0, p_1, p_2) \in \{(1/2, 0/1, 2), (0, 1/2, 1/2), (1/3, 1/3, 1/3)\}$ across all embedding dimensions and scenario types. However, when $(p_0, p_1, p_2) = (1/2, 1/2, 0)$, ICL already works well in the

*Table 3.* ICL on dual-*disconnected*-relationship tasks, averaged over 10 repetitions, achieves perfect accuracy when $(p_0, p_1, p_2) \in \{(1/2, 0/1, 2), (0, 1/2, 1/2), (1/3, 1/3, 1/3)\}$ regardless of architectures and embedding dimensions $(d_E)$. When $(p_0, p_1, p_2) = (1/2, 1/2, 0)$, ICL already performs well under the balanced scenario.

| | Balanced | | Imbalanced | | Extreme | |
|---|---|---|---|---|---|---|
| $(p_0, p_1, p_2)$ | $d_E = 10$ | $d_E = 100$ | $d_E = 10$ | $d_E = 100$ | $d_E = 10$ | $d_E = 100$ |
| $(0, 1, 0)$ | $(0, 0)$ | $(0, 0)$ | $(0, 0)$ | $(0, 0)$ | $(0, 0)$ | $(0, 0)$ |
| $(0, 0, 1)$ | $(0, 0)$ | $(0, 0)$ | $(0.16, 0.14)$ | $(0, 0)$ | $(0.21, 0.29)$ | $(0, 0)$ |
| $(1/2, 1/2, 0)$ | $(1, 1)$ | $(0.82, 0.83)$ | $(0.28, 0.27)$ | $(0.95, 0.95)$ | $(0.83, 0.85)$ | $(0.91, 0.91)$ |
| $(1/2, 0, 1/2)$ | $(1, 1)$ | $(1, 1)$ | $(1, 1)$ | $(1, 1)$ | $(1, 1)$ | $(1, 1)$ |
| $(0, 1/2, 1/2)$ | $(1, 1)$ | $(1, 1)$ | $(1, 1)$ | $(1, 1)$ | $(1, 1)$ | $(1, 1)$ |
| $(1/3, 1/3, 1/3)$ | $(1, 1)$ | $(1, 1)$ | $(1, 1)$ | $(1, 1)$ | $(1, 1)$ | $(1, 1)$ |

balanced scenario. This is because the two relationships are disjoint, thus making task selection easier. In addition, we consider a *contaminated* version of the training data where $cd$ (*ef*) sentences can contain some $e_i$'s and $f_i$'s ($c_i$'s and $d_i$'s). We also obtain a perfect accuracy when $(p_0, p_1, p_2) \in \{(1/2, 0/1, 2), (0, 1/2, 1/2), (1/3, 1/3, 1/3)\}$ across all embedding dimensions and scenario types.

### 2.4. Experiments on countries, states, and capital cities

We perform two experiments involving countries and their capital cities, as well as US states and their capital cities. Our prompts follow the format $c_1 d_1, c_2 d_2, \cdots, c_6 d_6, c_7$, where $c_i$ is a country or US state and $d_i$ is its capital city. Using LLaMA 2 (Touvron et al., 2023), we compare the prediction for each prompt with its corresponding $d_7$.

In the first experiment, we focus on 160 countries with a population exceeding one million in 2022. Among these countries, 31 have capital cities that are not their most populous cities, denoted by *type A*. The remaining 129 countries fall under *type B*. Each ICL prompt includes three type A countries among $c_1, \cdots, c_6$ to emphasize that the desired relationship is (country)–(capital) rather than (country)–(largest city). Subsequently, we randomly generate 1,000 prompts, with 500 having a $c_7$ representing a type A country and 500 having a $c_7$ representing a type B country. The ICL accuracies corresponding to type A and type B prompts are $0.58$ and $0.96$, respectively.

In the second experiment, we consider all 50 states (33 are of *type A* and 17 are of *type B*). The ICL accuracies corresponding to type A and type B prompts are $0.69$ and $0.84$, respectively. From both experiments, we notice that LLaMA 2 performs better on type B prompts (i.e., the capital city as the largest city). This suggests that ICL may arise from co-occurrence information, as larger cities tend to appear more frequently compared to smaller ones.

### 2.5. Experiments on a synthetic corpus

We conduct experiments on a synthetic corpus consisting of (country)–(capital) and (country)–(IOC

code) relationships. Each sentence in the corpus is categorized into exactly one of six possible categories: (1) exactly one country-capital pair; (2) exactly two country-capital pairs; (3) exactly one country-IOC pair; (4) exactly two country-IOC pairs; (5) exactly one country without any pair; and (6) no country. In sentences with country-capital pairs, each capital city can appear in any position relative to the country. Conversely, in sentences with country-IOC pairs, each IOC code must directly follow the country. See Appendix E for a detailed description of the corpus.

Two models are trained on this corpus: a CBOW and a five-layer two-head autoregressive transformer. Both models have an embedding dimension of 100. We then compare the ICL accuracies for both relationships given one to five in-context examples. For the CBOW model, the country-capital accuracies are $(0.81, 0.82, 0.78, 0.73, 0.65)$ and the country-IOC accuracies are $(0.15, 0.38, 0.59, 0.71, 0.79)$. Here, the $i$-th number corresponds to the accuracy given $i$ in-context examples. For the transformer, the accuracies are $(0.00, 0.15, 0.34, 0.22, 0.07)$ and $(1.00, 0.77, 0.78, 0.97, 0.99)$, respectively.

When using the transformer, we find that the accuracies for the country-IOC task are significantly higher compared to those for the country-capital task. This is likely because each IOC code consistently follows the corresponding country in the corpus, similar to ICL prompts. On the other hand, ICL fails to work on the country-capital task, where there is no consistent pattern in how each pair occurs in the corpus. Meanwhile, ICL works decently well on both tasks under the CBOW model.

## 3. The essential role of positional information in enabling in-context learning

In this section, we examine another common example of in-context learning (ICL), where the task involves predicting the first (or second) token given a sequence of tokens. To understand the significance of positional information (unlike the tasks in Section 2), we consider a simpler task: modeling

*Table 4.* Prediction accuracy with single/multi-layer models. For successful ICL, it is crucial that the first token of sentences in the training set covers the entire vocabulary (*Both*). Here, positional embeddings are essential, especially when using a one-layer model.

|  | Both | | Either | |
| --- | --- | --- | --- | --- |
| Pos. emb. | 1-layer | 5-layer | 1-layer | 5-layer |
| Learned | 1 | 1 | 0 | 0 |
| Sinusoidal | 1 | 1 | 0 | 0 |
| No pos. emb. | 0.30 | 0.89 | 0 | 0 |

sequences of tokens in the form $x_{i_1} x_{i_2} x_{i_3} x_{i_1}$. Theorem 3.1 underscores the necessity of incorporating positional information to correctly predict $x_{i_1}$ from $x_{i_1} x_{i_2} x_{i_3}$ in a single-layer model, and provides a construction of a basic attention-based model capable of achieving zero loss and perfect accuracy on this task. Its proof is in Appendix F.

**Theorem 3.1** (Necessity of modeling positions)**.** *Let the vocabulary be $\mathcal{V} = \{1, 2, \cdots, |V|\}$ and the training sequences take the form $x_{i_1} x_{i_2} x_{i_3} x_{i_1}$, where $x_{i_1} \neq x_{i_2} \neq x_{i_3} \neq x_{i_1}$ are chosen uniformly at random from $\mathcal{V}$. Consider a one-layer model that predicts the last $x_{i_1}$ via a learned function $f(\{x_{i_1}, x_{i_2}\}, x_{i_3})$ using the cross-entropy loss. In this case, it is not possible to achieve pefect accuracy or zero loss. On the other hand, we can achieve zero loss (and thus perfect accuracy) by incorporating positional information, i.e., via a learned function $\tilde{f}(\{(x_{i_1}, 1), (x_{i_2}, 2)\}, (x_{i_3}, 3))$.*

Here, $f(\{x_{i_1}, x_{i_2}\}, x_{i_3})$ represents a scenario where the model lacks positional information (e.g., $f$ is a one-layer autoregressive transformer without positional embeddings). Note that the output of this function is identical for inputs $x_{i_1} x_{i_2} x_{i_3}$ and $x_{i_2} x_{i_1} x_{i_3}$, which leads to the impossibility of attaining zero loss. In contrast, $\tilde{f}(\{(x_{i_1}, 1), (x_{i_2}, 2)\}, (x_{i_3}, 3))$ refers to a scenario where the model has access to positional information. We provide a construction of $\tilde{f}$ that achieves zero loss in Appendix F.

**Experiments.** We validate Theorem 3.1 by training transformers with causal masking to autoregressively learn sequences of the form $x_{i_1} x_{i_2} x_{i_3} x_{i_1}$, and assessing their accuracy in predicting the last token on a separate test data of the same pattern. We use $|V| = 20$ and an embedding dimension of 10. We consider these settings: (i) *number of layers*: 1, 5; (ii) *positional embeddings*: learned, sinusoidal, no positional embeddings; and (iii) *train-test split*: each token in the vocabulary is the first token in both the training and test sets (*Both*), each token in the vocabulary is the first token in either set, but not both (*Either*).

Table 4 summarizes the results. Two main findings emerge: (1) for the model to successfully generalize to unseen sentences, each token in $\mathcal{V}$ should be present as the first token in both the training and test sets; (2) positional embeddings

are crucial when using only one attention layer.

**Multiple layers.** With multi-layer models, positional information can be encoded without explicit positional embeddings. This is summarized in Proposition 3.2, whose proof is in Appendix G.

**Proposition 3.2** (Multi-layer models can encode positions)**.** *Consider the sentence $x_{i_1} x_{i_2} x_{i_3} x_{i_1}$. Using a two-layer autoregressive model, the model's final output for predicting the last $x_{i_1}$ is given by $t(x_{i_1} x_{i_2} x_{i_3}) := g_3\left(\{f_1(\{x_{i_1}\}), f_2(\{x_{i_1}\}, x_{i_2})\}, f_3(\{x_{i_1}, x_{i_2}\}, x_{i_3})\right)$ for some $f_1, f_2, f_3$, and $g_3$.*

Proposition 3.2 shows that we generally have $t(x_{i_1} x_{i_2} x_{i_3}) \neq t(x_{i_2} x_{i_1} x_{i_3})$, unlike in the one-layer case. Consequently, high accuracy is achievable without positional embeddings, as shown in Table 4. This result parallels findings in Haviv et al. (2022) that autoregressive transformers can implicitly encode positions.

**Roadmap of Section 3.** In the rest of this section, we consider settings where each sentence contains repeating patterns. Section 3.1 focuses on a simple scenario where training sentences follow the form *abacdc*, where $a \neq b$ and $c \neq d$, or a noisy variation of it. The ICL prompts maintain the same pattern but use different combinations of *ab* and *cd* from those in the training data. Our goal is to understand what types of training data facilitate ICL in clean or noisy scenarios. Section 3.2 explores a more realistic case where two possible patterns are present: repeating the first letter (*abca*) and repeating the second letter (*abcb*).

### 3.1. ICL on single-pattern tasks

In this section, we examine the case where the training sentences follow a specific pattern of the form *abacdc*. To replicate real-world training scenarios, we also analyze how incorporating nuisance tokens into the training sentences affects the ICL capability of autoregressive models. To formalize the discussion, let the vocabulary be $\mathcal{V} \cup \mathcal{N}$, where $\mathcal{N}$ represents the nuisance tokens. Define $S = \{(a, b) \mid a, b \in \mathcal{V}, a \neq b\}$ and partition $S$ into $S_1$ and $S_2$, where $\{c[1] \mid c \in S_1\} = \{c[1] \mid c \in S_2\} = \mathcal{V}$ and $c[i]$ denotes the $i$-th element of $c$, to ensure training sentences are distinct from the ICL prompts. Consider three different scenarios:

1. *Clean*: Training data follow the form *abacdc* where $ab, cd \in S_1$. ICL prompts follow the form abacd where ab, cd $\in S_2$.

2. *One-noisy*: Training data follow the form *abacdc* where $ab, cd \in S_1$, with one nuisance token $n \in \mathcal{N}$ randomly inserted anywhere except the last position (to ensure ICL prompts do not resemble the training data). ICL prompts follow the form abacd where ab, cd $\in S_2$.

3. *Block-noisy*: Training data follow the form *abacdc* where $ab, cd \in S_1$, with three consecutive nuisance tokens

$n_1, n_2, n_3 \in \mathcal{N}$ randomly inserted while preserving the *aba* and *cdc* blocks. ICL prompts follow the form *abacdcef* where $\underline{ab}$, $\underline{cd}$, $\underline{ef} \in S_2$.

We set the vocabulary size $|V| = 20$, the number of nuisance tokens $N = 20$, and use only one attention layer as we empirically showed that additional layers do not improve performance. Table 5 reveals interesting phenomena. Firstly, under the clean data scenario, ICL performs exceptionally well, with an observed performance increase with learned positional embeddings and a larger embedding dimension. However, ICL is notably challenging under the one-noisy scenario. In the block-noisy scenario, learned positional embeddings are crucial for satisfactory ICL performance. Theorem 3.3 formalizes these findings.

**Theorem 3.3** (Blocked nuisance token structure facilitates ICL). *Consider a sufficiently large autoregressive position-aware model that can achieve the minimum possible theoretical loss. Training this model in the one-noisy (block-noisy) scenario results in zero (perfect) ICL accuracy.*

The proof is in Appendix H. Theorem 3.3 says that ICL works perfectly under the block-noisy scenario, yet fails to work under the one-noisy scenario. However, as shown in Table 5, the use of sinusoidal positional embeddings significantly enhances prediction accuracy in the one-noisy scenario. This may be due to the fact that sinusoidal embeddings can encode relative positional information (Vaswani et al., 2017). For example, training sentences of the form *nabacdc*, where $n \in \mathcal{N}$, may help in predicting the most likely token following the ICL prompt $\underline{abacd}$.

### 3.2. ICL on dual-pattern tasks

We next examine the case where both the training data and ICL prompts contain two different patterns occurring with equal probability: *abcadefd* and *abcbdefe*, where $a \neq b \neq c \neq a$ and $d \neq e \neq f \neq d$. We consider the *clean* and *block-noisy* scenarios, defined similarly as in Section 3.1, and set $|V| = N = 20$. Table 6 outlines the ICL performance for both scenario types across different model configurations. Unlike the single-pattern scenario, there is a performance improvement with five layers compared to one layer, particularly with learned positional embeddings.

This phenomenon is related to the notion of induction heads, where at least two layers may be necessary to distinguish the two patterns (Olsson et al., 2022). This is reflected in Figure 2 in Appendix K, which compares the accuracy trajectories of one-layer and five-layer models. While the five-layer setup effectively differentiates the two patterns, the one-layer configuration fails to do so. Meanwhile, in both clean and block-noisy scenarios, learned positional embeddings lead to notably higher accuracies as compared to sinusoidal ones, similar to the single-pattern case.

## 4. Scenarios where ICL can fail

In this section, we consider two scenarios where in-context learning (ICL) can fail, irrespective of architectures. In Section 4.1, both the training data and test prompts follow repeating patterns across blocks, but the pattern in the test data differs from that in the training data. In Section 4.2, the training sentences contain known input-output pairs but only at fixed locations.

### 4.1. Failed scenario 1: Sentences with repeating patterns

In this scenario, our training data comprises sentences in the form of *abacdcefe*, where $a \neq b$, $c \neq d$, and $e \neq f$. Note that each sentence is composed of three blocks, each consisting of three tokens with the same pattern. For the ICL task, we consider predicting $\underline{f}$ from the prompt $\underline{abbcddef}$, where $\underline{a} \neq \underline{b}$, $\underline{c} \neq \underline{d}$, and $\underline{e} \neq \underline{f}$. As each training sentence contains a repeated pattern, we expect a well-trained model to output $\underline{f}$ to maintain the pattern seen in the in-context examples: $\underline{abb}$ and $\underline{cdd}$. However, as depicted in Table 7, all models fail to recognize it and predict the correct token.

We next formalize a generalization of this scenario. Let the vocabulary be $\mathcal{V} = \{1, 2, \cdots, |V|\}$, and define $S = \{(a, b) \mid a, b \in \mathcal{V}, a \neq b\}$. To ensure training sentences are distinct from the ICL prompts, we first partition $S$ into $S_1$ and $S_2$, where $\{c[1] \mid c \in S_1\} = \{c[1] \mid c \in S_2\} = \mathcal{V}$. Here, $c[i]$ denotes the $i$-th element of $c$. Suppose we autoregressively train a sufficiently large position-aware model so that it is possible to achieve the minimum possible theoretical loss. The training sentences take the form $x_{11}x_{12}x_{11}x_{21}x_{22}x_{21} \cdots x_{N1}x_{N2}x_{N1}$, where $x_{i1} \neq x_{i2}$ and $(x_{i1}, x_{i2})$ is independently selected from $S_1$ for every $i \in [N]$. Theorem 4.1 states that ICL fails to hold regardless of the number of in-context examples.

**Theorem 4.1** (Failure of ICL: Different repeated patterns). *Consider the generalized scenario in Section 4.1. For any $1 \leq \ell \leq N$, given an in-context prompt of the form $\underline{x_{11}x_{12}x_{12}x_{21}x_{22}x_{22}} \cdots \underline{x_{\ell 1}x_{\ell 2}}$ where $\underline{x_{i1}} \neq \underline{x_{i2}}$ and $(\underline{x_{i1}}, \underline{x_{i2}}) \in S_2$ for every $i \in [\ell]$, the model predicts $\underline{x_{\ell 1}}$ instead of $\underline{x_{\ell 2}}$ (Proof in Appendix I).*

Theorem 4.1 and Table 7 demonstrate that ICL achieves zero accuracy irrespective of the number of in-context examples ($\ell - 1$). This insight sheds light on the ICL capacity of autoregressive models. Simply put, if the pattern in the in-context examples differs significantly from any pattern in the training data, ICL may not occur. These results align with the findings of Raventós et al. (2023) and Yadlowsky et al. (2023) on the importance of data diversity for ICL.

### 4.2. Failed scenario 2: Sentences with known pairs but only at fixed locations

We revisit the paired relationship scenario discussed in Section 2. The training data now comprises sentences of the

*Table 5.* ICL on single-pattern tasks, averaged over 10 repetitions, achieves near-perfect accuracy in the clean data scenario regardless of architectures and embedding dimension ($d_E$). The one-noisy scenario is the most challenging, with sinusoidal embeddings giving a higher accuracy. In the block-noisy scenario, learned positional embeddings result in significantly better ICL performance.

| Pos. emb. | $d_E = 10$ | | | $d_E = 100$ | | |
|---|---|---|---|---|---|---|
| | Clean | One-noisy | Block-noisy | Clean | One-noisy | Block-noisy |
| Learned | 0.97 | 0.00 | 0.95 | 1.00 | 0.00 | 1.00 |
| Sinusoidal | 0.66 | 0.10 | 0.01 | 0.96 | 0.00 | 0.55 |
| RoPE (Su et al., 2024) | 0.31 | 0.00 | 0.03 | 0.48 | 0.00 | 0.00 |

*Table 6.* ICL on dual-pattern tasks, averaged over 10 repetitions, achieves notably better accuracy using learned than sinusoidal embeddings. Near-perfect accuracy is attained in the clean scenario by a 5-layer transformer with an embedding dimension ($d_E$) of 100 and learned positional embeddings. The block-noisy scenario is challenging; the same model attains the best performance.

| | Pos. emb. | $d_E = 10$ | | $d_E = 100$ | |
|---|---|---|---|---|---|
| | | Clean | Block-noisy | Clean | Block-noisy |
| 1-layer | Learned | (0.33, 0.33) | (0.15, 0.16) | (0.51, 0.49) | (0.49, 0.50) |
| | Sinusoidal | (0.12, 0.66) | (0.03, 0.03) | (0.51, 0.48) | (0.06, 0.10) |
| 5-layer | Learned | (0.39, 0.39) | (0.23, 0.22) | (0.97, 0.98) | (0.87, 0.70) |
| | Sinusoidal | (0.32, 0.34) | (0.04, 0.04) | (0.83, 0.82) | (0.04, 0.07) |

form of $a_i pqrs b_i$, where $(a_i, b_i)$ represents a known pairing and $p, q, r, s$ represent other words. For the ICL task, we consider predicting $b_{i_3}$ from the prompt $a_{i_1} b_{i_1} a_{i_2} b_{i_2} a_{i_3}$, where $i_1 \neq i_2 \neq i_3 \neq i_1$. As each training sentence always contains an $(a_i, b_i)$ pair at a fixed location, we expect a well-trained model to output $b_{i_3}$ to maintain the pattern in the in-context examples: $a_{i_1} b_{i_1}$ and $a_{i_2} b_{i_2}$. However, none of the models can identify the repeated patterns and predict the correct token, as shown in Table 7.

We next formalize a generalization of this scenario. Let the vocabulary be $\{(a_i, b_i)\}_{i \in [I]} \cup \mathcal{V}$, where $\mathcal{V} = \{1, 2, \cdots, |V|\}$ represent other words. As in Section 4.1, we autoregressively train a sufficiently large position-aware model that can achieve the minimum possible theoretical loss. The training sentences take the form $a_i v_1 v_2 \cdots v_{2k} b_i$, where $i$ and $v_{1:2k}$ are independently chosen from $[I]$ and $\mathcal{V}$, respectively, uniformly at random. Theorem 4.2, whose proof is in Appendix J, states that ICL fails to occur regardless of the number of in-context examples.

**Theorem 4.2** (Failure of ICL: Different pattern structures). *Consider the generalized scenario in Section 4.2. For any $1 \leq \ell \leq k + 1$, given an in-context prompt of the form $a_{i_1} b_{i_1} a_{i_2} b_{i_2} \cdots a_{i_\ell}$ with distinct $i_j$'s, the model never predicts $b_{i_\ell}$: it predicts a uniform probability vector over $\mathcal{V}$ when $1 \leq \ell \leq k$, and $b_{i_1}$ when $\ell = k + 1$.*

Theorem 4.2 highlights that the success of ICL relies heavily on how the patterns appear in the training data. In this scenario, the $(a_i, b_i)$ pairs consistently appear at the beginning and end of each training sentence, and we anticipate

*Table 7.* ICL in failed scenarios, averaged over 10 repetitions, achieves zero accuracy for any architecture and embedding dimension ($d_E$).

| | Pos. emb. | Failed scenario 1 | | Failed scenario 2 | |
|---|---|---|---|---|---|
| | | $d_E = 10$ | $d_E = 100$ | $d_E = 10$ | $d_E = 100$ |
| 1-layer | Learned | 0.00 | 0.00 | 0.01 | 0.00 |
| | Sinusoidal | 0.01 | 0.00 | 0.00 | 0.00 |
| 5-layer | Learned | 0.00 | 0.00 | 0.00 | 0.00 |
| | Sinusoidal | 0.00 | 0.00 | 0.00 | 0.00 |

the model to recognize this relationship for ICL to occur. However, as shown in Theorem 4.2 and Table 7, this is not the case. An empirical study is provided in Appendix L.

## 5. Discussion

This paper investigates how in-context learning (ICL) can arise from pretraining on unstructured natural language data. We present three main findings, supported by both theory and empirical studies. First, ICL can be achieved by simply modeling co-occurrence using classical language models like continuous bag of words (CBOW), when ICL prompts involve pairs that frequently appear together. Second, when ICL prompts involve recognizable patterns that do not always co-occur, positional information and nuisance token structure play crucial roles in enabling ICL. Finally, we highlight the importance of training data structure in ICL by examining two instances where ICL can fail. Further analyses on other ICL tasks and their reliance on model architecture can be fruitful avenues for future work.

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

# A. Related work

Large language models (LLMs), such as transformers, are widely recognized for their outstanding performance in in-context learning (ICL) (Brown et al., 2020). ICL refers to the capability of LLMs to discern specific tasks and generate predictions based on input-output pairs (known as prompts) without needing any parameter updates. A multitude of studies have been dedicated to exploring this intriguing phenomenon from various theoretical and empirical perspectives. In this section, we provide a brief summary of some of these studies.

Some studies adopted a Bayesian approach to studying ICL. Xie et al. (2021) posited that ICL can be viewed as implicit Bayesian inference. They demonstrated that LLMs can infer a latent document-level concept for next-token prediction during pretraining and a shared latent concept across input-output pairs in an ICL prompt, under the assumption that documents are generated from hidden Markov models (HMMs). Wang et al. (2023) and Zhang et al. (2023) expanded on this idea by exploring more realistic latent variable models beyond HMMs. Wang et al. (2023) argued that large language models function as latent variable models, with latent variables containing task-related information being implicitly inferred. Zhang et al. (2023) showed that without updating the neural network parameters, ICL can be interpreted as Bayesian model averaging parameterized by the attention mechanism. Panwar et al. (2023) provided empirical evidence that transformers behave like Bayesian predictors when performing ICL with linear and non-linear function classes. Dalal and Misra (2024) proposed a Bayesian learning framework to understand ICL through the lens of text generation models represented by multinomial transition probability matrices. Chiang and Yogatama (2024) proposed the pelican soup framework to explain ICL without relying on latent variable models. This framework incorporates concepts such as a common sense knowledge base, natural language classification, and meaning association, enabling the establishment of a loss bound for ICL that depends on the number of in-context examples.

Garg et al. (2022) formulated ICL as learning a specific function class $\mathcal{F}$ from prompts of the form $(x_1, f(x_1), \ldots, x_n, f(x_n), x_{n+1})$ and their corresponding responses $f(x_{n+1})$. Here, $f \in \mathcal{F}$, where $\mathcal{F}$ is a function class. In this context, ICL refers to the capability of a transformer to output a number close to $g(y_{n+1})$ given a prompt of the form $(y_1, g(y_1), \ldots, y_n, g(x_n), y_{n+1})$, where $g \in \mathcal{F}$. Many studies adopted this regression formulation of ICL, with some linking ICL to gradient descent. Akyürek et al. (2022); Von Oswald et al. (2023), and Dai et al. (2023) proved that transformers are capable of implementing gradient descent, which results in their ICL ability. Bai et al. (2023) established generalization bounds for ICL and proved that transformers can perform algorithm selection like statisticians. Zhang et al. (2024) showed that the gradient flow dynamics of transformers converge to a global minimum that enables ICL. Huang et al. (2023) investigated the learning dynamics of single-layer softmax transformers trained via gradient descent to perform ICL on linear functions. Ahn et al. (2024) explored the optimization landscape of transformers and proved that the optimal parameters coincide with an iteration of preconditioned gradient descent.

In a related exploration, Li et al. (2023a) showed that softmax regression models learned through gradient descent are similar to transformers. Ren and Liu (2023) related ICL with softmax transformers to contrastive learning, where the inference process of ICL can be viewed as a form of gradient descent. Mahankali et al. (2023) proved that minimizing the pretraining loss is equivalent to a step of gradient descent in single-layer linear transformers. Vladymyrov et al. (2024) established that linear transformers execute a variant of preconditioned gradient descent by maintaining implicit linear models. On the other hand, some studies argued that the ICL ability of transformers cannot be attributed to gradient descent. Fu et al. (2023) showed that ICL for linear regression tasks arises from higher-order optimization techniques like iterative Newton's method rather than gradient descent. Wibisono and Wang (2023) demonstrated that transformers can perform ICL on unstructured data that lack explicit input-output pairings, with softmax attention playing an important role especially when using a single attention layer. Shen et al. (2023) provided empirical evidence that the equivalence between gradient descent and ICL might not be applicable in real-world scenarios. In contrast to these studies, our work provides a connection between ICL and classical language models like continuous bag of words (CBOW). Specifically, we show that ICL can arise by modeling co-occurrence patterns via CBOW.

Numerous studies focused on the pretraining aspects (e.g., data distribution and task diversity) of ICL. Min et al. (2022) showed that the input-label mapping in the in-context examples does not significantly affect ICL performance. Chan et al. (2022) demonstrated that the ICL capabilities of transformers depend on the training data distributions and model features. Kossen et al. (2024) established that ICL considers in-context label information and is capable of learning entirely new tasks in-context. Li and Qiu (2023) introduced an iterative algorithm designed to enhance ICL performance by selecting a small set of informative examples that effectively characterize the ICL task. Qin et al. (2023) proposed a method based on zero-shot chain-of-thought reasoning for selecting ICL examples, emphasizing the importance of choosing diverse

examples that are strongly correlated with the test sample. Han et al. (2023b) studied ICL by identifying a small subset of the pretraining data that support ICL via gradient-based methods. They discovered that this supportive pretraining data typically consist of more uncommon tokens and challenging examples, characterized by a small information gain from long-range context. Peng et al. (2024) proposed a selection method for ICL demonstrations that are both data-dependent and model-dependent. Van et al. (2024) introduced a demonstration selection method that enhances ICL performance by analyzing the influences of training samples using influence functions.

In a similar vein, Wu et al. (2023) demonstrated that pretraining single-layer linear attention models for ICL on linear regression with a Gaussian prior can be effectively accomplished with a minimal number of independent tasks, regardless of task dimension. Raventós et al. (2023) emphasized a task diversity threshold that differentiates the conditions under which transformers can successfully address unseen tasks. Yadlowsky et al. (2023) attributed the impressive ICL capabilities of transformers to the diversity and range of data mixtures in their pretraining, rather than their inductive biases for generalizing to new tasks. Ding et al. (2024) compared the ICL performance of transformers trained with prefixLM (where in-context samples can attend to all tokens) versus causalLM (where in-context samples cannot attend to subsequent tokens), finding that the latter resulted in poorer ICL performance. Chen et al. (2024b) discovered that the ICL capabilities of language models rely on the presence of pairs of phrases with similar structures within the same sentence. Zhao et al. (2024) proposed a calibration scheme that modifies model parameters by adding random noises, resulting in fairer and more confident predictions. Abbas et al. (2024) demonstrated that the ICL predictions from transformer-based models often exhibit low confidence, as indicated by high Shannon entropy. To address this issue, they introduced a straightforward method that linearly calibrates output probabilities, independent of the model's weights or architecture. Similar to these works, our work highlights the importance of training data structure for ICL to arise.

Other studies analyzed ICL from a learning theory perspective. Hahn and Goyal (2023) proposed an information-theoretic bound that explains how ICL emerges from next-token prediction. Wies et al. (2023) derived a PAC-type framework for ICL and finite-sample complexity results. Jeon et al. (2024) introduced a novel information-theoretic view of meta-learning (including ICL), allowing for the decomposition of errors into three components. They proved that in ICL, the errors decrease as the number of examples or sequence length increase. Other studies focus on the mechanistic interpretability component of ICL. Olsson et al. (2022) argued that transformers can develop induction heads that are able to complete token sequences such as [A][B] $\cdots$ [A] $\rightarrow$ [B], leading to impressive ICL performance. Bietti et al. (2023) examined a setup where tokens are generated from either global or context-specific bigram distributions to distinguish between global and in-context learning. They found that global learning occurs rapidly, while in-context learning is achieved gradually through the development of an induction head. Ren et al. (2024) identified semantic induction heads that increase the output logits of tail tokens when attending to head tokens, providing evidence that these heads could play a vital role in the emergence of ICL. Yu and Ananiadou (2024) showed that the ICL ability of transformers arises from the utilization of in-context heads, where each query and key matrix collaborate to learn the similarity between the input text and each demonstration example.

A number of works delved into specific data generating processes to provide insight into the emergence of ICL. Bhattamishra et al. (2023) examined the ICL ability of transformers by focusing on discrete functions. Specifically, they showed that transformers perform well on simpler tasks, struggle with more complex tasks, and can learn more efficiently when provided with examples that uniquely identify a task. Guo et al. (2023) investigated ICL in scenarios where each label is influenced by the input through a potentially complex yet constant representation function, coupled with a unique linear function for each instance. Akyürek et al. (2024) studied ICL of regular languages produced by random finite automata. They compared numerous neural sequence models and demonstrated that transformers significantly outperform RNN-based models because of their ability to develop *n-gram heads*, which are a generalization of *induction heads*. Sander et al. (2024) analyzed simple first-order autoregressive processes to gain insight into how transformers perform ICL to predict the next tokens. Our work focuses on data generating processes containing input-output relationship pairs or repeating token patterns to better understand the importance of co-occurrence, positional information, and training data structure for ICL.

Some studies explored how different components of transformers affect their ICL abilities. Ahuja and Lopez-Paz (2023) compared the ICL performance of transformers and MLP-based architectures under distribution shifts. Their findings demonstrate that while both methods perform well in in-distribution ICL, transformers exhibit superior ICL performance when faced with mild distribution shifts. Collins et al. (2024) showed that softmax attention outperforms linear attention in ICL due to its ability to calibrate its attention window to the Lipschitzness of the pretraining tasks. Xing et al. (2024) focused on linear regression tasks to identify transformer components that enable ICL. They found that positional encoding is crucial, along with the use of multiple heads, multiple layers, and larger input dimensions. Cui et al. (2024) proved that multi-head attention outperforms single-head attention in various practical scenarios, including those with noisy labels and

correlated features. Chen et al. (2024a) investigated the ICL dynamics of a multi-head softmax attention model applied to multi-task linear regression. They proved the convergence of the gradient flow and observed the emergence of a *task allocation* phenomenon, where each attention head specializes in a specific task.

Finally, several studies proposed various hypotheses on the emergence of ICL and provided theoretical justifications. Swaminathan et al. (2023) introduced clone-structured causal graphs (CSCGs) to explain how ICL can generalize to unseen sentences via a mechanism called rebinding. Li et al. (2023b) viewed ICL as an algorithm learning problem where a transformer implicitly constructs a hypothesis function at inference time. Han et al. (2023a) argued that the ability of transformers to execute ICL is attributable to their capacity to simulate kernel regression. Singh et al. (2023) explored the interaction between ICL and in-weights learning (IWL) using synthetic data designed to support both processes. They observed that ICL initially emerges, followed by a transient phase where it disappears and gives rise to IWL. Yan et al. (2023) studied ICL from the perspective that token co-occurrences play a crucial role in guiding the learning of surface patterns that facilitates ICL. Abernethy et al. (2024) showed that transformers can execute ICL by dividing a prompt into examples and labels, then employing sparse linear regression to deduce input-output relationships and generate predictions. Lin and Lee (2024) developed a probabilistic model that can simultaneously explain both task learning and task retrieval aspects of ICL. Here, task learning refers to the ability of language models to identify a task from in-context examples, while task retrieval pertains to their ability to locate the relevant task within the pretraining data.

# B. Proof of Theorem 2.1

*Proof.* Let $|V| = 2K + L$ denote the vocabulary size. Consider a sentence $X$ represented by its one-hot encoding (i.e., $X \in \{0, 1\}^{|V| \times S}$). For every position $i \in [S]$, the loss for predicting the word in the $i$-th position given all the other words is given by $\|AX(\mathbb{1}_S - e_i) - Xe_i\|_2^2$, where $A = \frac{U^\top V}{S-1} \in \mathbb{R}^{|V| \times |V|}$ and $e_i \in \mathbb{R}^S$ is a zero vector with 1 on its $i$-th entry. Here, $U$ ($V$) is a matrix consisting of the center (context) embeddings of all tokens, and $A$ is a matrix summarizing the similarity between each pair of words (one as a center word and the other as a context word). Our objective is to find $A$ that minimizes the sum of losses for each position in each sentence. Lemma B.1 gives a closed-form expression of the minimizer.

**Lemma B.1.** *The minimizer of the overall loss is given by $A = B\left((S - 2)B + C\right)^{-1}$. Here, $B$ is a matrix whose $(i, j)$-th entry is $p(i, j)$, the probability that for a given (center, context) pair, the center is $i \in |V|$ and the context is $j \in |V|$. Moreover, $C$ is a diagonal matrix whose $i$-th diagonal entry is $p(i) = \sum_{j \in |V|} p(i, j)$.*

*Proof.* Let $\mathcal{L}(X) = \sum_{i=1}^S \|AX(\mathbb{1}_S - e_i) - Xe_i\|_2^2$ denote the sum of the losses corresponding to all tokens in sentence $X$. By direct calculation,

$$\frac{\partial \mathcal{L}(X)}{\partial A} = 2AX \left(\sum_{i=1}^S (\mathbb{1}_S - e_i)(\mathbb{1}_S - e_i)^\top\right) X^\top - 2X \left(\sum_{i=1}^S e_i(\mathbb{1}_S - e_i)^\top\right) X^\top$$

.

Note that $\sum_{i=1}^S (\mathbb{1}_S - e_i)(\mathbb{1}_S - e_i)^\top = (S - 2)\mathbb{1}_{S \times S} + \mathbb{I}_{S \times S}$ and $\sum_{i=1}^S e_i(\mathbb{1}_S - e_i)^\top = \mathbb{1}_{S \times S} - \mathbb{I}_{S \times S}$. Now, let our sentences be $X_1, X_2, \cdots, X_N$. The minimizer of the overall loss thus satisfies

$$A \frac{1}{N} \sum_{k=1}^N X_k \left((S - 2)\mathbb{1}_{S \times S} + \mathbb{I}_{S \times S}\right) X_k^\top = \frac{1}{N} \sum_{k=1}^N X_k \left(\mathbb{1}_{S \times S} - \mathbb{I}_{S \times S}\right) X_k^\top. \tag{1}$$

We denote the number of (center, context) pairs across all sentences in which the center is $i$ and the context is $j$ by $\#(i, j)$. Moreover, we define $\#(i) = \sum_{j \in |V|} \#(i, j)$. It is easy to see that Equation (1) can be rewritten as

$$A \left((S - 2)\tilde{B} + \tilde{C}\right) = \tilde{B},$$

where $\tilde{B}$ is a matrix such that its $(i, j)$-th entry is $\frac{\#(i,j)}{N}$ and $\tilde{C}$ is a diagonal matrix such that its $i$-th diagonal element is $\frac{\#(i)}{N}$. As $N \to \infty$, an application of the law of large numbers yields $\frac{\#(i,j)}{N} \to S(S - 1)p(i, j)$ almost surely and $\frac{\#(i)}{N} \to S(S - 1)p(i)$ almost surely, where $p(i, j)$ is the probability that for a given (center, context) pair, the center is $i$ and the context is $j$, and $p(i) = \sum_{j \in |V|} p(i, j)$.

Thus, as $N \to \infty$, we have

$$A = B\left((S - 2)B + C\right)^{-1},$$

where $B$ and $C$ are defined in the statement of Lemma B.1. $\qquad \square$

We now define

- $p_1 = p(c_i, c_j) = p(d_i, d_j) = p(c_i, d_j) = p(d_i, c_j)$ for any $i \neq j$;

- $p_2 = p(r_i, r_j)$ for any $i \neq j$;

- $p_3 = p(c_i, d_i) = p(d_i, c_i)$ for any $i$;

- $p_4 = p(c_i, r_j) = p(d_i, r_j) = p(r_j, c_i) = p(r_j, d_i)$ for any $i, j$,

where the equalities in the probabilities are a consequence of the data distribution.

For ease of presentation, we denote a square matrix with $\alpha$ on the diagonal and $\beta$ off the diagonal as $X_{\alpha,\beta}$, and a matrix with all entries $\gamma$ as $Y_\gamma$. We then have

$$B = \begin{bmatrix} X_{0,p_1} & X_{p_3,p_1} & Y_{p_4} \\ X_{p_3,p_1} & X_{0,p_1} & Y_{p_4} \\ Y_{p_4} & Y_{p_4} & X_{0,p_2} \end{bmatrix}.$$

Now, define $a = (S-2)p_1$, $b = (S-2)p_2$, $c = (S-2)p_3$, $d = (S-2)p_4$, $e = 2(K-1)p_1 + p_3 + Lp_4$, and $f = (L-1)p_2 + 2Kp_4$. It is easy to see that

$$(S-2)B + C = \begin{bmatrix} X_{e,a} & X_{c,a} & Y_d \\ X_{c,a} & X_{e,a} & Y_d \\ Y_d & Y_d & X_{f,b} \end{bmatrix}.$$

Moreover, its inverse can be written as

$$((S-2)B + C)^{-1} = \begin{bmatrix} X_{q_5,q_1} & X_{q_3,q_1} & Y_{q_4} \\ X_{q_3,q_1} & X_{q_5,q_1} & Y_{q_4} \\ Y_{q_4} & Y_{q_4} & X_{q_6,q_2} \end{bmatrix},$$

where

$\Delta = 2a(K-1)(b(L-1) + f) + b(L-1)(c+e) + cf - 2d^2KL + ef,$

$q_1 = -\left( \frac{-abL + ab - af + d^2L}{(2a-c-e)\Delta} \right),$

$q_2 = \frac{2ab(K-1) + b(c+e) - 2d^2K}{(b-f)\Delta},$

$q_3 = -\left( \frac{\begin{array}{c} -2a^2b(K-1)(L-1) - 2a^2f(K-1) + 2abc(K-2)(L-1) + 2acf(K-2) \\ + 2(a-c)d^2KL + bc(c+e)(L-1) + cf(c+e) + d^2L(c-e) \end{array}}{(c-e)(2a-c-e)\Delta} \right),$

$q_4 = -\left( \frac{d}{\Delta} \right),$

$q_5 = -\left( \frac{\begin{array}{c} -2a^2b(K-1)(L-1) - 2a^2f(K-1) + 2abe(K-2)(L-1) + 2aef(K-2) \\ + 2(a-e)d^2KL + be(c+e)(L-1) + ef(c+e) + d^2L(e-c) \end{array}}{(e-c)(2a-c-e)\Delta} \right),$

and $q_6 = -\left( \frac{2a(K-1)(b(L-2)+f) + b(L-2)(c+e) + cf - 2d^2KL + 2d^2K + ef}{(b-f)\Delta} \right).$

By computing $A = B((S-2)B + C)^{-1}$, given the following center words, the similarities between them and all possible context words are as follows:

- Center word $= c_i$ for any $i$

    - $c_i : 2(K-1)p_1q_1 + p_3q_3 + Lp_4q_4;$
    - $c_j : 2(K-2)p_1q_1 + p_1q_5 + p_3q_1 + p_1q_3 + Lp_4q_4 \ (j \neq i);$
    - $d_i : 2(K-1)p_1q_1 + p_3q_5 + Lp_4q_4;$
    - $d_j : 2(K-2)p_1q_1 + p_1q_3 + p_3q_1 + p_1q_5 + Lp_4q_4 \ (j \neq i);$
    - $r_j : 2(K-1)p_1q_4 + p_3q_4 + p_4q_6 + (L-1)p_4q_2 \ \text{(for any } j).$

- Center word $= d_i$ for any $i$

  – $d_i : 2(K-1)p_1q_1 + p_3q_3 + Lp_4q_4$;
  – $d_j : 2(K-2)p_1q_1 + p_1q_5 + p_3q_1 + p_1q_3 + Lp_4q_4$ $(j \neq i)$;
  – $c_i : 2(K-1)p_1q_1 + p_3q_5 + Lp_4q_4$;
  – $c_j : 2(K-2)p_1q_1 + p_1q_3 + p_3q_1 + p_1q_5 + Lp_4q_4$ $(j \neq i)$;
  – $r_j : 2(K-1)p_1q_4 + p_3q_4 + p_4q_6 + (L-1)p_4q_2$ (for any $j$).

- Center word = $r_i$

  – $c_j : 2(K-1)p_4q_1 + p_4q_5 + p_4q_3 + (L-1)p_2q_4$ (for any $j$);
  – $d_j : 2(K-1)p_4q_1 + p_4q_5 + p_4q_3 + (L-1)p_2q_4$ (for any $j$);
  – $r_i : 2Kp_4q_4 + (L-1)p_2q_2$;
  – $r_j : 2Kp_4q_4 + (L-2)p_2q_2 + p_2q_6$ $(j \neq i)$.

Recall that the ICL problem of interest is the following: given context words $c_{i_1} d_{i_1} \cdots c_{i_\ell} d_{i_\ell} c_{i_{\ell+1}}$, we aim to predict $d_{i_{\ell+1}}$. Without loss of generality, we can rewrite the problem to predict $d_{\ell+1}$ given context words $c_1 d_1 \cdots c_\ell d_\ell c_{\ell+1}$. We now compute the total similarity for each possible center word, where $\epsilon^\top \delta$ indicates the similarity between the word $\epsilon$ in the center and the word $\delta$ in the context.

- $c_1$ (or any of $c_2, \cdots, c_\ell$) : $c_1^\top c_1 + \ell c_1^\top c_2 + c_1^\top d_1 + (\ell-1)c_1^\top d_2$;

- $d_1$ (or any of $d_2, \cdots, d_\ell$) : $c_1^\top d_1 + \ell c_1^\top d_2 + c_1^\top c_1 + (\ell-1)c_1^\top c_2$;

- $r_1$ (or any other $r_k$'s) : $(\ell+1)r_1^\top c_1 + \ell r_1^\top d_1 = (2\ell+1)r_1^\top c_1$;

- $c_{\ell+1}$ : $\ell c_1^\top c_2 + \ell c_1^\top d_2 + c_1^\top c_1$;

- $d_{\ell+1}$ : $\ell c_1^\top d_2 + \ell c_1^\top c_2 + c_1^\top d_1$;

- $c_{\ell+2}$ (or any $c_k$'s not in the context prompt) : $(\ell+1)c_1^\top c_2 + \ell c_1^\top d_2$;

- $d_{\ell+2}$ (or any $d_k$'s not in the context prompt) : $(\ell+1)c_1^\top d_2 + \ell c_1^\top c_2$.

Note that correctly predicting $d_{\ell+1}$ is equivalent to the following conditions being simultaneously satisfied:

- $c_1^\top d_1 > c_1^\top c_1$, equivalent to $p_3q_5 > p_3q_3$;

- $c_1^\top d_2 > c_1^\top c_1$ and $c_1^\top c_2 > c_1^\top c_1$, equivalent to $p_1q_3 + p_3q_1 + p_1q_5 > 2p_1q_1 + p_3q_3$;

- $c_1^\top d_1 > c_1^\top c_2$ and $c_1^\top d_1 > c_1^\top d_2$, equivalent to $2p_1q_1 + p_3q_5 \geq p_1q_5 + p_1q_3 + p_3q_1$;

- $2\ell c_1^\top c_2 + c_1^\top d_1 > (2\ell+1)r_1^\top c_1$, equivalent to $2\ell(2(K-2)p_1q_1 + p_1q_5 + p_3q_1 + p_1q_3 + Lp_4q_4) + 2(K-1)p_1q_1 + p_3q_5 + Lp_4q_4 > (2\ell+1)(2(K-1)p_4q_1 + p_4q_5 + p_4q_3 + (L-1)p_2q_4)$;

In our data generating process, it is easy to see that $p_1 = 0$, $p_2 = \frac{(S-2)(S-3)}{L(L-1)}$, $p_3 = \frac{1}{K}$, and $p_4 = \frac{S-2}{KL}$, where each $p_i$ is multiplied by a constant $S(S-1) > 0$ (without loss of generalization) to make calculations easier. From here, we have $a = 0$, $b = \frac{(S-2)^2(S-3)}{L(L-1)}$, $c = \frac{S-2}{K}$, $d = \frac{(S-2)^2}{KL}$, $e = \frac{S-1}{K}$, and $f = \frac{(S-1)(S-2)}{L}$. Substituting to the above, we have

- $q_1 = \frac{(S-2)^4}{\Delta KL(2S-3)}$;

- $q_3 = \frac{-K(S-2)^2(S-1)^2 - (S-2)^4}{\Delta KL(2S-3)}$;

- $q_4 = \frac{-(2S-3)(S-2)^2}{\Delta KL(2S-3)}$;

- $q_5 = \frac{K(S-2)(S-1)^3 + (S-2)^4}{\Delta KL(2S-3)}$,

where $\Delta = \frac{(S-1)^2(S-2)}{KL} > 0$.

We now check when these conditions are simultaneously satisfied. The first condition is equivalent to $p_3 > 0$ and $K > \frac{2(S-2)^3}{(S-1)^2(2S-3)}$, which always hold. The second condition reduces to $p_3 > 0$ and $2(S-2)^4 + K(S-2)^2(S-1)^2 > 0$, which is also true. The third condition can be written as $p_3 > 0$ and $K(S-2)(S-1)^3 > 0$, which always hold. The last condition becomes

$$(2\ell + 1)((K+L)(S-2)^2(S-1) + K(S-2)(S-1)^2 - 2(S-2)^4) < KL(S-1)^3,$$

which is equivalent to

$$2\ell + 1 < \frac{KL(S-1)^3}{(K+L)(S-2)^2(S-1) + K(S-2)(S-1)^2 - 2(S-2)^4}.$$

Note that this condition ensures that the model predicts $d_{\ell+1}$ instead of one of the $r_i$'s. $\qquad\square$

## C. Comparison of ICL performance using squared and cross-entropy loss across different numbers of examples

*Table 8.* ICL performance in the *clean* scenario, evaluated with both squared and cross-entropy loss functions across different numbers of examples (0 to 8) with $d_E = 100$, averaged over 10 repetitions.

| $(p_0, p_1, p_2)$ | Squared | | | | | Cross-entropy | | | | |
|---|---|---|---|---|---|---|---|---|---|---|
| | 0 | 2 | 4 | 6 | 8 | 0 | 2 | 4 | 6 | 8 |
| $(0, 1, 0)$ | 1 | 1 | 0 | 0 | 0 | 0.87 | 0 | 0 | 0 | 0 |
| $(0, 0, 1)$ | 1 | 1 | 1 | 0 | 0 | 1 | 0 | 0 | 0 | 0 |
| $(1/2, 1/2, 0)$ | 1 | 1 | 1 | 1 | 1 | 1 | 1 | 0.34 | 0 | 0 |
| $(1/2, 0, 1/2)$ | 1 | 1 | 1 | 1 | 1 | 1 | 1 | 1 | 1 | 1 |
| $(0, 1/2, 1/2)$ | 1 | 1 | 1 | 1 | 1 | 1 | 1 | 1 | 0 | 0 |
| $(1/3, 1/3, 1/3)$ | 1 | 1 | 1 | 1 | 1 | 1 | 1 | 1 | 1 | 0 |

*Table 9.* ICL performance in the *corrupted* scenario, evaluated with both squared and cross-entropy loss functions across different numbers of examples (0 to 8) with $d_E = 100$, averaged over 10 repetitions.

| $(p_0, p_1, p_2)$ | Squared | | | | | Cross-entropy | | | | |
|---|---|---|---|---|---|---|---|---|---|---|
| | 0 | 2 | 4 | 6 | 8 | 0 | 2 | 4 | 6 | 8 |
| $(0, 1, 0)$ | 1 | 0 | 0 | 0 | 0 | 0 | 0 | 0 | 0 | 0 |
| $(0, 0, 1)$ | 1 | 0.97 | 0 | 0 | 0 | 1 | 0 | 0 | 0 | 0 |
| $(1/2, 1/2, 0)$ | 1 | 1 | 1 | 0.53 | 0 | 1 | 0 | 0 | 0 | 0 |
| $(1/2, 0, 1/2)$ | 1 | 1 | 1 | 1 | 1 | 1 | 1 | 1 | 1 | 1 |
| $(0, 1/2, 1/2)$ | 1 | 1 | 0.76 | 0 | 0 | 1 | 1 | 0 | 0 | 0 |
| $(1/3, 1/3, 1/3)$ | 1 | 1 | 1 | 1 | 1 | 1 | 1 | 1 | 0.18 | 0 |

From Tables 8 and 9, we observe that ICL with CBOW on single-relationship tasks performs better with squared loss compared to cross-entropy loss and with fewer demonstration examples. Also, ICL tends to deteriorate after a certain number of in-context demonstrations. As detailed in Appendix B, a smaller number of examples (e.g., zero) allows the model to produce the correct output instead of one of the $r_i$'s. This is in contrast with transformer-based LLMs, which achieve better ICL performance as the number of demonstrations increases. On the other hand, ICL on dual-connected-relationship tasks requires at least one demonstration example. On the other hand, ICL on dual-relationship tasks as described in Section 2.2 requires at least one demonstration example to distinguish between the two tasks.

# D. Proof of Theorem 2.2

*Proof.* We show that given a prompt of the form $c_{i_1} d_{i_1} \cdots c_{i_\ell} d_{i_\ell} c_{i_{\ell+1}}$ with distinct $i_k$'s, a trained CBOW model is more likely to predict $d_{i_{\ell+1}}$ than $e_{i_{\ell+1}}$. If this is established, the other part of the theorem follows analogously. We now define

- $p_1 = p(c_i, d_j) = p(d_i, c_j) = p(d_i, d_j) = p(c_i, e_j) = p(e_i, c_j) = p(e_i, e_j)$ for any $i \neq j$;

- $p_2 = p(r_i, r_j)$ for any $i \neq j$;

- $p_3 = p(c_i, d_i) = p(d_i, c_i) = p(c_i, e_i) = p(e_i, c_i)$;

- $p_4 = p(d_i, r_j) = p(r_i, d_j) = p(e_i, r_j) = p(r_i, e_j)$ for any $i, j$;

where the equalities in the probabilities are a consequence of the data distribution. By direct calculation, we have $p_1 = \frac{1}{K(K-1)}$, $p_2 = \frac{(S-4)(S-5)}{L(L-1)}$, $p_3 = \frac{1}{K}$, and $p_4 = \frac{S-4}{KL}$, where each $p_i$ is multiplied by $S(S-1) > 0$ (without loss of generality) to make calculations easier. Moreover, it is easy to see that $p(c_i, r_j) = p(r_i, c_j) = 2p_4$ for any $i, j$ and $p(c_i, c_j) = 2p_1$ for any $i \neq j$. Lastly, we define $a = (S-2)p_1$, $b = (S-2)p_2$, $c = (S-2)p_3$, $d = (S-2)p_4$, $e = 2(K-1)p_1 + p_3 + Lp_4$, and $f = 4Kp_4 + (L-1)p_2$.

The next step the proof is to use Lemma B.1 in Appendix B to obtain the similarity matrix $A$. As previously, we denote a square matrix with $\alpha$ on the diagonal and $\beta$ off the diagonal as $X_{\alpha,\beta}$, and a matrix with all entries $\gamma$ as $Y_\gamma$. We then have

$$B = \begin{bmatrix} X_{0,2p_1} & X_{p_3,p_1} & X_{p_3,p_1} & Y_{2p_4} \\ X_{p_3,p_1} & X_{0,p_1} & Y_0 & Y_{p_4} \\ X_{p_3,p_1} & Y_0 & X_{0,p_1} & Y_{p_4} \\ Y_{2p_4} & Y_{p_4} & Y_{p_4} & X_{0,p_2} \end{bmatrix}$$

and

$$(S-2)B + C = \begin{bmatrix} X_{2e,2a} & X_{c,a} & X_{c,a} & Y_{2d} \\ X_{c,a} & X_{e,a} & Y_0 & Y_d \\ X_{c,a} & Y_0 & X_{e,a} & Y_d \\ Y_{2d} & Y_d & Y_d & X_{f,b} \end{bmatrix}. \tag{2}$$

Moreover, its inverse can be written as

$$((S-2)B + C)^{-1} = \begin{bmatrix} X_{q_2,q_1} & X_{q_3,q_1} & X_{q_3,q_1} & Y_{q_4} \\ X_{q_3,q_1} & X_{q_5,q_6} & X_{q_7,q_8} & Y_{q_4} \\ X_{q_3,q_1} & X_{q_7,q_8} & X_{q_5,q_6} & Y_{q_4} \\ Y_{q_4} & Y_{q_4} & Y_{q_4} & X_{q_9,q_{10}} \end{bmatrix}, \tag{3}$$

for some $q_1, q_2, \cdots, q_{10}$. Recall that our task is show that given context words $c_{i_1}, d_{i_1}, \cdots, c_{i_\ell}, d_{i_\ell} c_{i_{\ell+1}}$ with distinct $i_k$'s, the center word is more likely to be $d_{i_{\ell+1}}$ than $e_{i_{\ell+1}}$. In other words, we need to establish that

$$d_{i_{\ell+1}}^\top c_{i_1} + d_{i_{\ell+1}}^\top d_{i_1} + \cdots + d_{i_{\ell+1}}^\top c_{i_\ell} + d_{i_{\ell+1}}^\top d_{i_\ell} + d_{i_{\ell+1}}^\top c_{i_\ell+1}$$
$$> e_{i_{\ell+1}}^\top c_{i_1} + e_{i_{\ell+1}}^\top d_{i_1} + \cdots + e_{i_{\ell+1}}^\top c_{i_\ell} + e_{i_{\ell+1}}^\top d_{i_\ell} + e_{i_{\ell+1}}^\top c_{i_\ell+1},$$

where $\epsilon^\top \delta$ indicates the similarity between the word $\epsilon$ in the center and the word $\delta$ in the context. This similarity can be obtained from the matrix $A = B((S-2)B + C)^{-1}$. By symmetry, the inequality reduces to $d_i^\top d_j > e_i^\top d_j$ for any $i \neq j$.

By computing the matrix $A$, we have

$$d_i^\top d_j = p_3 q_1 + p_1 q_3 + (K-2)p_1 q_1 + (K-2)p_1 q_6 + Lp_4 q_4 + p_1 q_5$$

and

$$e_i^\top d_j = p_3 q_1 + p_1 q_3 + (K-2)p_1 q_1 + (K-2)p_1 q_8 + p_1 q_7 + Lp_4 q_4.$$

Thus, our problem again reduces to showing $(K-2)q_6 + q_5 > (K-2)q_8 + q_7$ as $p_1 = \frac{1}{K(K-1)} > 0$. Upon multiplying (3) and (2) and equating the result with the identity matrix, we have the following equations:

$$a(K-1)q_1 + cq_3 + dLq_4 + eq_5 + a(K-1)q_6 = 1 \tag{4}$$
$$(c + a(K-2))q_1 + aq_3 + dLq_4 + aq_5 + (e + a(K-2))q_6 = 0 \tag{5}$$
$$a(K-1)q_1 + cq_3 + dLq_4 + eq_7 + a(K-1)q_8 = 0 \tag{6}$$
$$(c + a(K-2))q_1 + aq_3 + dLq_4 + aq_7 + (e + a(K-2))q_8 = 0. \tag{7}$$

Comparing (5) and (7) yields

$$a(((K-2)q_6 + q_5) - ((K-2)q_8 + q_7)) = e(q_8 - q_6).$$

As $a = (S-2)p_1 > 0$ and $e = 2p_1(K-1) + p_3 + p_4L > 0$, we now only need to show that $q_8 > q_6$. Comparing (4) and (6) as well as (5) and (7), we have

$$a(q_5 - q_7) = (e + a(K-2))(q_8 - q_6)$$
$$e(q_5 - q_7) = a(K-1)(q_8 - q_6) + 1,$$

which reduces to $(q_8 - q_6)(e^2 + ae(K-2) - a^2(K-1)) = a$. The conclusion follows since $a > 0$ and

$$e^2 + ae(K-2) - a^2(K-1) = (e-a)(e + a(K-1)) = \left(\frac{S-1}{K} - \frac{S-2}{K(K-1)}\right)(e + a(K-1)) > 0.$$

$\square$

## E. Corpus generation process for experiments in Section 2.5

1. Randomly select 10 countries and obtain their capital cities and IOC codes.

2. Generate 30 sentences containing exactly one country-capital pair (3 for each country).
   *Example: Paramaribo is the vibrant heart of Suriname.*

3. Generate 30 sentences containing exactly one country-IOC pair (3 for each country).
   *Example: Gabon (GAB) protects its diverse rainforests and wildlife.*

4. Generate 30 sentences containing exactly one country without any pair.
   *Example: The banking sector is central to Liechtenstein's prosperity.*

5. Generate 60 sentences without any country, capital city, or IOC code.
   *Example: Every country has its unique cultural identity and heritage.*

6. Generate 810 sentences containing exactly two different country-capital pairs by concatenating sentences generated in Step 2.
   *Example: The city of Dushanbe reflects Tajikistan's vibrant spirit. Roseau is the cultural tapestry of Dominica.*

7. Generate 810 sentences containing exactly two different country-IOC pairs by concatenating sentences generated in Step 3.
   *Example: Mayotte (MAY) features lush landscapes and peaks. Turkmenistan (TKM) features the fiery Darvaza Crater.*

## F. Proof of Theorem 3.1

*Proof.* Consider the instance of predicting $a$ from $abc$, i.e., $f(\{a, b\}, c)$. By the assumption on the data distribution, it is equally likely that the task is predicting $b$ from $bac$. In this case, the corresponding function is also $f(\{a, b\}, c)$. Thus, the sum of the cross-entropy losses corresponding to these two tasks is lower bounded by $2 \log 2 > 0$. Also, it is easy to see that we cannot achieve perfect accuracy since the predictions for $abc$ and $bac$ must be the same.

We now show that it is possible to attain zero loss and perfect accuracy when the model includes positional embeddings, so that $\tilde{f}(\{(a, 1), (b, 2)\}, (c, 3)) \neq \tilde{f}(\{(b, 1), (a, 2)\}, (c, 3))$. As a special case, we consider a simplified version of the transformer architecture, where

$$\tilde{f}(\{(a, 1), (b, 2)\}, (c, 3)) = \frac{\sum_{k \in \{a,b,c\}} (x_k + p_1) \exp((x_k + p_1)^\top (x_c + p_3))}{\sum_{k \in \{a,b,c\}} \exp((x_k + p_1)^\top (x_c + p_3))}.$$

and

$$p(d \mid abc) \propto \exp\left(x_d^\top \tilde{f}(\{(a, 1), (b, 2)\}, (c, 3))\right).$$

for any token $d$. Here, $x_i$ and $p_j$ represent the embedding of token $i$ and position $j$, respectively.

Let $p_1^\top p_3 = p$, $p_2^\top p_3 = q$, $p_3^\top p_3 = r$, $x_i^\top x_i = s$, $x_i^\top x_j = t$ for any $i \neq j$, $p_1^\top x_i = u$ for any $i$, $p_2^\top x_i = v$ for any $i$, and $p_3^\top x_i = w$ for any $i$. Note that this holds due to the assumed data generating process. We consider the following construction: $p_1 = b\mathbb{1}_{|V|}$, $p_2 = p_3 = \mathbb{1}_{|V|}$, and $x_i = ae_i$, where $e_i$ is a zero vector with 1 on the $i$-th entry. This implies $p = b|V|$, $q = r = |V|$, $s = a^2$, $t = 0$, $u = ab$, and $v = w = a$.

By direct calculation, the cross-entropy loss of predicting $a$ from $abc$ is given by

$$-\log\left(\frac{\exp(\alpha_1 a^2)}{\exp(\alpha_1 a^2) + \exp(\alpha_2 a^2) + \exp(\alpha_3 a^2) + |V| - 3}\right),$$

where

$$\alpha_1 = \frac{\exp(ab + b|V|)}{\exp(ab + b|V|) + \exp(a + |V|) + \exp(a^2 + a + |V|)},$$

$$\alpha_2 = \frac{\exp(a + |V|)}{\exp(ab + b|V|) + \exp(a + |V|) + \exp(a^2 + a + |V|)},$$

$$\alpha_3 = \frac{\exp(a^2 + a + |V|)}{\exp(ab + b|V|) + \exp(a + |V|) + \exp(a^2 + a + |V|)}.$$

Letting $b = a^2$ and $a \to \infty$, it is easy to see that we can bring the cross-entropy loss arbitrarily close to zero. Consequently, we also have a perfect prediction accuracy. $\square$

## G. Proof of Proposition 3.2

*Proof.* The intermediate representation of the first layer is given by $f_1(\{x_{i_1}\})$, $f_2(\{x_{i_1}\}, x_{i_2})\}$, $f_3(\{x_{i_1}, x_{i_2}\}, x_{i_3})$, and $f_4(\{x_{i_1}, x_{i_2}, x_{i_3}\}, x_{i_1})$, for some functions $f_1, f_2, f_3$, and $f_4$. To predict the last $x_{i_1}$, we use the third coordinate of the second layer representation, which is given by $t(x_{i_1} x_{i_2} x_{i_3}) := g_3(\{f_1(\{x_{i_1}\}), f_2(\{x_{i_1}\}, x_{i_2})\}, f_3(\{x_{i_1}, x_{i_2}\}, x_{i_3}))$, for some function $g_3$. It is easy to see that in general, $t(x_{i_1} x_{i_2} x_{i_3}) \neq t(x_{i_2} x_{i_1} x_{i_3})$. $\square$

## H. Proof of Theorem 3.3

*Proof.* In the one-noisy scenario, each sentence takes one of the following forms: *nabacdc*, *anbacdc*, *abnacdc*, *abancdc*, *abacndc*, and *abacdnc*, where $n \in \mathcal{N}$. In order to achieve the minimum possible theoretical loss, we minimize each loss term separately. Concretely, the minimum loss of predicting the sixth token given the first five tokens is attained by the following rule:

- When the first five tokens do not contain any nuisance token, output a uniform probability vector over $\mathcal{N}$.

- Otherwise, output the conditional probability of $c[2]$ given $x$, where $(x, c[2]) \in S_1$. Here, $x$ represents the last non-nuisance token.

Under this rule, the predicted output for any in-context example *abacd* is never $c$, since $c \notin \mathcal{N}$. In the block-noisy scenario, each sentence takes one of the following forms: $n_1 n_2 n_3 abacdc$, $aban_1 n_2 n_3 cdc$, and $abacdcn_1 n_2 n_3$, where $n_1, n_2, n_3 \in \mathcal{N}$. The minimum loss of predicting the ninth token given the first eight tokens is attained by the following rule:

- When the seventh token is not a nuisance token, output the seventh token with probability one.

- When the seventh token is a nuisance token, output a uniform probability vector over $\mathcal{N}$.

Under this rule, the predicted output for any in-context example *abacdcef* is $e$, resulting in perfect ICL accuracy. $\square$

## I. Proof of Theorem 4.1

*Proof.* Recall that each training sentence is of the form $x_{11} x_{12} x_{11} x_{21} x_{22} x_{21} \cdots x_{N1} x_{N2} x_{N1}$. Note that we can decompose the total loss $\mathcal{L}$ into $\mathcal{L}_1 + \mathcal{L}_2 + \cdots + \mathcal{L}_{3N}$, where $\mathcal{L}_g$ denotes the loss of predicting the $g$-th token given all the other previous tokens. As the $x_{i1} x_{i2} x_{i1}$ blocks are generated independently, the optimal loss should satisfy $\mathcal{L}_1 = \mathcal{L}_4 = \cdots = \mathcal{L}_{[3N-2]} = \mathcal{L}_{[1]}$, $\mathcal{L}_2 = \mathcal{L}_5 = \cdots = \mathcal{L}_{[3N-1]} = \mathcal{L}_{[2]}$, and $\mathcal{L}_3 = \mathcal{L}_6 = \cdots = \mathcal{L}_{[3N]} = \mathcal{L}_{[3]}$. Therefore, it is sufficient to minimize $\mathcal{L}_{[1]} + \mathcal{L}_{[2]} + \mathcal{L}_{[3]}$.

In order to achieve the minimum possible theoretical loss, we need to minimize $\mathcal{L}_{[1]}$, $\mathcal{L}_{[2]}$, and $\mathcal{L}_{[3]}$ separately. It is easy to see that $\mathcal{L}_{[1]}$ is minimized by outputting the marginal probability of $c[1]$, where $c \in S_1$. Similarly, $\mathcal{L}_{[2]}$ is minimized by outputting the conditional probability of $c[2]$ given $x_{i_1}$, where $(x_{i_1}, c[2]) \in S_1$. On the other hand, it is possible to achieve an $\mathcal{L}_{[3]}$ value of zero by outputting $x_{i_1}$ with probability one.

Now, given an ICL prompt $\underline{x_{11} x_{12} x_{12} x_{21} x_{22} x_{22} \cdots x_{\ell 1} x_{\ell 2}}$ where $\ell \leq N$, the trained model should predict $x_{\ell 1}$ with probability one since $\{c[1] \mid c \in S_2\} = \mathcal{V}$ and our ICL prompt corresponds to $\mathcal{L}_{[3]}$. This completes the proof. $\square$

## J. Proof of Theorem 4.2

*Proof.* We proceed similarly as the proof of Theorem 4.1. Concretely, we separately minimize $\mathcal{L}_g$ for $g \in [2k+2]$, where $\mathcal{L}_g$ denotes the loss of predicting the $g$-th token given all the other previous tokens. It is easy to see that $\mathcal{L}_1$ is minimized by outputting a uniform probability vector over $a_{1:[I]}$, whereas $\mathcal{L}_h$ (for any $2 \leq h \leq 2k+1$) is minimized by outputting a uniform probability vector over $\mathcal{V}$. Moreover, it is possible to achieve an $\mathcal{L}_{2k+2}$ value of zero by outputting $b_i$ with probability one.

From here, given an ICL prompt of the form $a_{i_1} b_{i_1} a_{i_2} b_{i_2} \cdots a_{i_\ell}$, the trained model should predict a uniform probability vector over $\mathcal{V}$ if $\ell \leq k$, and $b_{i_1}$ if $\ell = k+1$. In all cases, the model does not predict $b_{i_\ell}$, completing the proof. $\square$

## K. Distinguishing two different patterns requires more than one layers

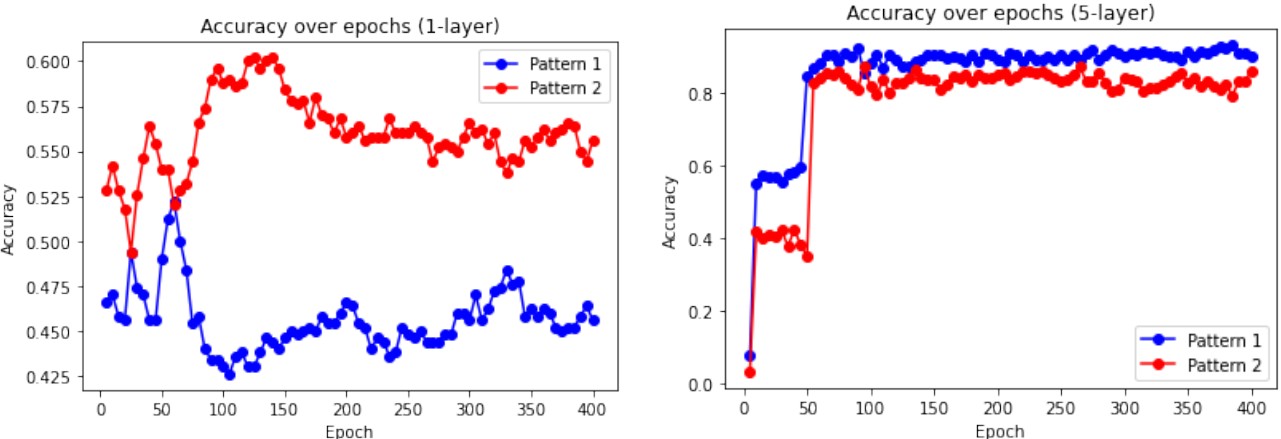

*Figure 2.* One-layer models fail to differentiate the two patterns in Section 3.2, as evidenced by the accuracy trajectory graph on the left. On the other hand, five-layer models are capable of doing so.

## L. Experiments for Section 4.2

We conduct an experiment on a synthetic corpus consisting of *(country)-(capital)* relationships. Each sentence in the corpus is categorized into exactly one of four possible categories: (1) exactly one country-capital pair; (2) exactly two country-capital pairs; (3) exactly one country without any pair; and (4) no country. In sentences with exactly one country-capital pairs, each capital appears in the first position, each country appears in the last position, and every sentence consists of six words (similar to the setting in Section 4.2). The corpus generation process is as follows:

1. Randomly select 10 countries and obtain their capital cities and IOC codes.

2. Generate 130 sentences containing exactly one country-capital pair (13 for each country). *Example: Paramaribo stands as capital of Suriname.*

3. Generate 30 sentences containing exactly one country without any pair.
   *Example: The banking sector is central to Liechtenstein's prosperity.*

4. Generate 60 sentences without any country, capital city, or IOC code.
   *Example: Every country has its unique cultural identity and heritage.*

5. Generate 1,000 sentences containing exactly two different country-capital pairs by concatenating sentences generated in Step 2.
   *Example: Brazil functions as heart of Brasilia. Turkmenistan operates as center for Ashgabat.*

We train a five-layer two-head autoregressive transformer on this corpus, with an embedding dimension of 100. Similar to Section 2.5, we assess the ICL accuracies using prompts involving countries and their capitals. We discover that the ICL accuracies are zero regardless of the number of in-context examples (one to five), thus supporting the theory.

## M. Limitations of this work

This study has several limitations. Firstly, the experiments are conducted on a relatively small scale. However, they still provide sufficient evidence to support the theoretical findings. Secondly, the focus of this study is on two specific types of in-context learning (ICL) tasks, as described in Section 1. Lastly, real data sets are not utilized due to the lack of alignment with the study objectives. Despite these limitations, we believe that this work offers valuable insights into how ICL arises through training on unstructured natural language data, supported by both theoretical and empirical evidence from experiments involving prompting and synthetic data. Further analyses on other ICL tasks and their reliance on model architecture can be fruitful avenues for future work.

# N. Details of experiments and data sets

All experiments utilize the Keras package in Python, employing the Adam optimizer (Kingma and Ba, 2015) with a learning rate of 0.01. Early stopping is applied based on validation loss with a patience threshold of 5, utilizing a randomly selected subset representing 50% of the original data set. Each transformer layer uses two heads, as we empirically demonstrated that increasing the number of heads does not impact performance in our experiments. Each layer consists of the following components (in order): (1) Keras' multi-head causal self-attention block, with key_dim = value_dim = embed_dim/2; (2) Skip connection and layer normalization; (3) One hidden layer feed-forward network using the ReLU activation with dimension $= 2\times$ embed_dim; and (4) Skip connection and layer normalization.

The `world_population.csv` data set, used for the experiments in Sections 2.4 and 2.5, is obtained from `https://www.kaggle.com/datasets/iamsouravbanerjee/world-population-dataset`. According to the author, this data set is created from `https://worldpopulationreview.com/`.

The `us-state-capitals.csv` data set, used for the experiments in Section 2.4, is obtained from `https://github.com/jasperdebie/VisInfo/blob/master/us-state-capitals.csv`. Its source is unclear.

The `uscities.csv` data set, used for the experiments in Section 2.4, is obtained from `https://simplemaps.com/data/us-cities`, with a CC 4.0 license.

