# OpenReview forum: "In-Context Learning from Training on Unstructured Data: The Role of Co-Occurrence, Positional Information, and Training Data Structure"
_ICML.cc/2024/Workshop/ICL — ICML 2024 Workshop ICL Poster_

### Official Review · Reviewer_ThAp · 2024-06-08
**Interesting probe setup, but requires more thorough analysis and justification**

**Rating:** 1
**Fit:** 3
**Confidence:** 2

**Workshop Review:**

The assumptions used in the theorems seem somewhat contrived and not necessarily relevant to practice. It is also not clear how critical such assumptions might be for the deriving similar qualitative behavior. The problem setup for the empirical analysis is interesting and might possibly result in interesting insights, but I am of the opinion that it would require more thorough justification and possibly further work.

**Reason For Not Giving Higher Score:**

For the following reasons (comments, and questions unaddressed in the paper)
- Why should the number of ICL examples not be too large in Theorem 2.1?
- Zero accuracy (in the first two rows of table 1) with cross-entropy loss seems surprising/concerning. It would be interesting to find any discernible pattern in example failures.
- The assumptions behind Theorem 2.2 seem too contrived to be of relevance to a practical setting.
- Prior works demonstrate the essential role of "concepts (Xie et. al. on Bayesian ICL) or "schemas" (Swaminathan et. al., [ref]) for ICL. It would be interesting to see fleshed out any conceptual relation / contrast between this and a CBOW model.
- Swaminathan et. al. [ref] also demonstrate the role of "rebinding" in how ICL can generalize to handle examples unseen at training time. IIUC the current CBOW model ignores this aspect; which might merit an explicit mention in that case.

[ref]: https://proceedings.neurips.cc/paper_files/paper/2023/hash/5bc3356e0fa1753fff7e8d6628e71b22-Abstract-Conference.html

PS: Some other suggestions for improvement
- I think there's a subscript formatting bug in fig 1 (section 2, right column)
- It would be worth adding to the appendix a section on the model hyperparameters & training details. These details (burstiness of the training distribution, choice of curriculum, and transient nature of ICL) would influence the observations strongly.

**Reason For Not Giving Lower Score:**

The paper considers an important question, and takes an interesting angle of attack. It has the potential to result in valuable insights if the analysis & justifications were more careful/thorough.

---

### Official Review · Reviewer_wTGM · 2024-06-08
**Interesting settings and observations but flawed execution**

**Rating:** 2
**Fit:** 3
**Confidence:** 2

**Workshop Review:**

This submission studies how in-context learning (ICL) emerges from training on data that does not contain explicit in-context learning sequences.
Considering simple synthetic data generation processes, the authors highlight two important factors: co-occurrence patterns and positional patterns.
Regarding co-occurence patterns, they prove and provide experimental evidence that in some settings continuous bag of words (CBOW) models can complete patterns like `country1 capital1 country2 capital2 country3 ?` when trained on sequences containing countries, captials, and other tokens in arbitrary order.
Regarding positional patterns, they prove that autoregressive transformers that are not permutation invariant (with causal masking or explicit positional information) can use stable positional patters to complete unseen sequences of a seen task at test time (a weaker form of ICL).

## Significance and novelty
This work addresses an important and highly relevant topic -- emergence of in-context learning from training on unstructured text. Related work is cited properly. The main result that CBOW models can do in-context learning relying only on co-occurence information is interesting and novel to my best knowledge. However, this result explains little about emergence of ICL in modern LLMs. One indication of this is that ICL in CBOW models works until a certain number of demonstrations is reached (Theorem 2.1), which is unlike LLMs that benefit from increasing the number of in-context examples.
Furthermore, there is some evidence in Section 2.5 showing that number of context examples has *negative* effect on CBOW ICL performance, which is again not the case of LLMs.

The results related to importance of modeling positional information and helpfulness of stable positional patterns for ICL are also interesting.
It is likely that some types of ICL in modern LLMs exploit positional patterns.
However, Proposition 3.2 which shows that causal transformers with no explicity positional embedding mechanism can model positional information is not novel (see for example, [1] and [2]).

## Clarity
This submission is a bit hard to read, but becomes clearer upon rereading. Please see my additional comments below.

## Correctness
* Experimental results of Section 2.1 do not agree perfectly with Theorem 2.1 (similarly experiments corresponding to Theorem 2.2). The authors hypothesize that the difference in loss function (MSE in theory and cross-entropy in experiments) is the culprit. I would have loved to see experiments with MSE loss validating correctness of Theorem 2.1. Also, it would have been great to validate that CBOW ICL breaks after certain number of in-context examples.
* Theorems 3.3, 4.1, and 4.2 are wrong. These 3 theorems are similar to each other as they state that a sufficiently large transformer well-trained on a particular data will not be able to complete a certain pattern at test time. The common flaw in the proofs is that the authors assume that the learned solution is unique. One could construct transformers that have zero training loss but implement a special algorithm "on the side" to handle ICL-type sequences.
* I did not verify the proofs of Theorems 2.1 and 2.2.

## Minor comments/questions
* Figure 1. It would be helpful to highlight the pattern in Section 3 and Section 4 data. Also can be helpful to indicate the correct output. For Section 4 part, it would be informative to additionally indicate what wrong answer is predicted.
* The phrase "noise structure" does not refer to anything specific outside of the scope of this paper. It would be better to not use such generic terms. Furthermore, it seems that what is important in Section 3 experiments is that noise tokens to not break solutions that rely on positional patterns (especially absolute position patterns).
* Theorems should be stated clearly and be as self-contained as possible.
* It would be helpful add an explanation on why CBOW ICL learning in Theorem 2.1 works only until certain number of demonstrations.
* Theorem 2.1: If the co-occurence information is the primary driver of ICL, would be helpful to find out what happens with zero context examples (i.e., in the zero-shot setting).
* Theorem 2.1: I suggest to simplify the context length bound or replaced it with a maybe more conservative but simpler bound.
* Looking at the experimental results presented in Table 1, a natural question arises -- why does ICL fail when all input sequences contain exactly one pair, but works when half of the sequences contain one pair while the other half contain no pairs? I am referring to the "clean" case (0, 1, 0) vs (1/2, 1/2, 0).
* Lines 160-162, left part, "Moreover, the same analysis applies when each sentence consists of exactly two (instead of one)
different (ci, di) pairs". I suggest to expand on this. This does not seem trivial to me.
* Theorem 2.2 does not tell whether ICL works or not. Is it hard to derive the correctness of ICL too?
* Why is not the number of demonstrations a concern in Theorem 2.2 as in Theorem 2.1?
* There should be more details on how sentences are generated exactly in the experiments of Section 2.5.
* In Section 2.5, the difference between CBOW ICL accuracies for "country-captial" and "country-IOC code" sequences needs to be explained. Should not they be almost the same, provided that the only difference in the data generation process is order-related (countries and capitals appear in any order, while IOC codes follow countries).
* Lines 311-321, left side: I suggest to present more details about these experiments. I assume that causal masking was used. In that case it is expected that a 5-layer transformer can model positions.
* Line 355, left side: As far as I can tell, the experiments are not computationally expensive. Therefore, it would be more appropriate to consider transformers with at least a few layers. As authors noted themselves, some in-context learning mechanisms might need require than 1 transformer layers. Similarly, I suggest using more than 2 attention heads per layer.
* Lines 373-378, left side: I suggest trying RoPE.
* The ICL failure presented in Section 4.1 is not really surprising, because with only one task in training set one cannot expect to generalize to other tasks. The authors cited a few works highlighting diversity of training tasks.

Since there is going to be no rebuttal, please interpret the questions above as suggestions on how to improve this work.


## References
[1] Haviv A, Ram O, Press O, Izsak P, Levy O. Transformer language models without positional encodings still learn positional information. arXiv:2203.16634.
[2] Kazemnejad A, Padhi I, Natesan Ramamurthy K, Das P, Reddy S. The impact of positional encoding on length generalization in transformers. NeurIPS 2023.

**Reason For Not Giving Higher Score:**

Concerns about correctness of theoretical results

**Reason For Not Giving Lower Score:**

The results related to ICL in CBOW models are interesting and can be useful for the community.

---

### Meta-Review · Area_Chair_t8Fv · 2024-06-17

**Recommendation:** 2

**Metareview:**

Reviewers agree that this paper targets interesting questions on dataset effects on ICL, and provides an important side result that CBOWs can suffice for ICL. Reviewers also questioned some of the assumptions put in place to derive theoretical results and the correctness of some theorems, which the authors should address in the camera-ready.

---

### Decision · Program_Chairs · 2024-06-17

**Decision:**

Accept (Poster)

**Comment:**

**Accept with minor revision**: Please address the correctness concerns of both reviewers.